# Proximal Supervised Fine-Tuning

**Wenhong Zhu**[1,2]  **Ruobing Xie**[3,*]  **Rui Wang**[1,2,*]  **Xingwu Sun**[3,4]  **Di Wang**[3]  **Pengfei Liu**[1,2,*]
[1]Shanghai Jiao Tong University    [2]Shanghai Innovation Institute
[3]Large Language Model Department, Tencent    [4]University of Macau
{zwhong714, wangrui12, pengfei}@sjtu.edu.cn
{xrbsnowing}@163.com

## Abstract

Supervised fine-tuning (SFT) of foundation models often leads to poor generalization, where prior capabilities deteriorate after tuning on specific tasks. Inspired by trust-region policy optimization (TRPO) and proximal policy optimization (PPO) in reinforcement learning (RL), we propose `Proximal SFT (PSFT)`, a fine-tuning objective that incorporates the benefits of trust-region, effectively constraining policy drift during SFT while maintaining competitive tuning. By viewing SFT as a special case of policy gradient methods with constant positive advantages, we derive `PSFT` that **stabilizes optimization** and **leads to generalization**, while **leaving room for further optimization in subsequent post-training stages**. Experiments across mathematical, human-value, and multimodal domains show that PSFT matches standard SFT in-domain, outperforms it in out-of-domain generalization, remains stable under prolonged training without causing entropy collapse, and provides a stronger foundation for the subsequent optimization.

## 1 Introduction

Recently, post-training has become a crucial part of the overall training process. In particular, reinforcement learning (RL) algorithms, such as Proximal Policy Optimization (PPO) (Schulman et al., 2017) and Group Relative Policy Optimization (GRPO) (Shao et al., 2024), have demonstrated significant effectiveness when applied to language models (LMs) focused on reasoning tasks. As RL is scaled over time, foundation models gain the capacity to address complex problems through more profound and extended reasoning (OpenAI, 2024; Guo et al., 2025; Zhu et al., 2025), producing high-quality reasoning trajectories. Numerous community efforts have focused on leveraging this knowledge via supervised fine-tuning (SFT) (Guha et al., 2025; Li et al., 2025), a distillation approach valued for its efficiency and simplicity compared to RL.

However, SFT models are often criticized for poor generalization (Huan et al., 2025). This limitation arises because SFT essentially performs behavior cloning, which can result in weak generalization when the fine-tuning dataset is suboptimal or distributionally misaligned with the pretraining data (Chu et al., 2025), potentially causing large policy updates (Schulman et al., 2015). RL fine-tuning (RFT) provides a promising alternative, as numerous studies have shown that RL better preserves the generalization capabilities that SFT tends to erode (Chu et al., 2025; Huan et al., 2025).

Another concern is maintaining the ability to explore. In practice, SFT is often used as a cold start to stabilize RFT training. The gains from RFT may largely come from refining capabilities already acquired during pre-training and SFT (Gandhi et al., 2025). However, excessive reliance on SFT can diminish a model's capacity for exploration (Xie et al., 2024), as it would cause the entropy collapse (Cui et al., 2025) and thereby constrain exploration during RL training (Yu et al., 2025).

Therefore, the current challenges lie in improving the **generalization** and **exploration** abilities of SFT models. To tackle these challenges, this paper proposes an improved fine-tuning strategy `Proximal Supervised Fine-Tuning (PSFT)` that avoids rote learning and achieves reliable performance in post-training. Specifically, we establish a theoretical connection between SFT and RL, and then introduce a novel objective based on a clipped surrogate objective from PPO to SFT, which leverages the benefits of trust regions to constrain policy updates.

---

[*]Corresponding authors.

Our experiments demonstrate that, compared to standard SFT, PSFT attains comparable performance on the target task while preserving the model's general capabilities. Moreover, PSFT mitigates entropy collapse during training and yields superior target and generalization performance in subsequent RL stages. We evaluate PSFT on mathematics, human value alignment, and multimodal tasks, highlighting its broad applicability in practice. Our contributions are as follows:

- We propose PSFT, a novel SFT optimization method that utilizes a clipped surrogate objective to enforce trust-region-like constraints, preventing excessive policy updates. Compared to standard SFT, it preserves the model's general capabilities while achieving comparable performance on target tasks.
- PSFT effectively prevents entropy collapse and overfitting during SFT, thereby enabling subsequent RL stages to achieve more robust and superior results on both target and general tasks.
- We extensively validate PSFT across various base models, tasks, settings, and evaluation metrics, demonstrating its potential as a promising alternative to standard SFT.

## 2 PRELIMINARIES AND MOTIVATIONS

We formulate autoregressive language modeling as a Markov Decision Process (MDP) $(\mathcal{S}, \mathcal{A}, P, R)$. Here, $\mathcal{S}$ denotes the state space ( the prefix of the current decoding step), $\mathcal{A}$ the action space (possible next tokens), and $P(s' \mid s, a)$ the transition probability $\pi_\theta$, and $R(s, a)$ the reward function. Given a query $x = (x_1, \ldots, x_m)$, the initial state is $s_0 = x$. A response $y = (y_1, \ldots, y_n)$ is generated by sequentially selecting actions $a_t \in \mathcal{A}$. The policy defines the probability of $y$ given $x$ as

$$\pi_\theta(y \mid x) = \prod_{t=1}^{n} \pi_\theta(a_t \mid s_t), \quad s_t = (x, y_{<t}), \ y_{<t} = (y_1, \ldots, y_{t-1}). \tag{1}$$

### 2.1 SFT AS THE SPECIAL POLICY UPDATE

**Supervised Fine-tuning.** The training objective of SFT is to minimize the cross-entropy loss between the model's predicted token distribution and the ground truth tokens, which is defined as:

$$L^{\text{SFT}}(\theta) = -\hat{\mathbb{E}}_{(s_t, a_t^*) \sim \mathcal{D}} \left[ \log \pi_\theta(a_t^* \mid s_t) \right], \tag{2}$$

where $(s_t, a_t^*)$ pairs are sampled from an offline dataset $\mathcal{D}$, with $a_t^*$ representing the ground-truth token.

**Policy Gradient.** In contrast, policy gradient directly samples trajectories from the current policy $\pi_\theta$ as it interacts with the environment. Using the policy gradient theorem (Sutton et al., 2000), the objective is to maximize the expected return:

$$L^{\text{PG}}(\theta) = \hat{\mathbb{E}}_{(s_t, a_t) \sim \pi_\theta} \left[ \log \pi_\theta(a_t \mid s_t) \, \hat{A}_t \right], \tag{3}$$

where $\hat{A}_t$ is the estimated advantage function at step $t$, which measures how much better an action $a_t$ is compared to the average action at state $s_t$ under the current policy.

From this perspective, SFT can be seen as a special case of policy gradient where the sampling is from a fixed offline dataset $\mathcal{D}$, and the advantage is fixed as $\hat{A}_t = 1$ for the ground-truth actions, which corresponds to maximizing the likelihood of ground-truth tokens.

### 2.2 TRUST-REGION CONSTRAINTS IN RL

**Trust Region Policy Optimization.** TRPO (Schulman et al., 2015) maximizes a surrogate objective by leveraging trajectories collected under the old policy $\pi_{\theta_{\text{old}}}$, introducing an importance sampling ratio $r_t(\theta)$ to reweight these samples for evaluating the new policy $\pi_\theta$:

$$L^{\text{CPI}}(\theta) = \hat{\mathbb{E}}_{(s_t, a_t) \sim \pi_{\theta_{\text{old}}}} \left[ \frac{\pi_\theta(a_t \mid s_t)}{\pi_{\theta_{\text{old}}}(a_t \mid s_t)} \hat{A}_t \right] = \hat{\mathbb{E}}_t \left[ r_t(\theta) \hat{A}_t \right], \tag{4}$$

where the superscript *CPI* refers to conservative policy iteration (Kakade & Langford, 2002). TRPO ensures that each policy update stays within a trust region by enforcing a hard KL constraint, preventing the new policy from deviating too far from the previous one and thus maintaining stable and reliable learning.

**Proximal Policy Optimization.** However, directly maximizing $L^{\text{CPI}}$ with a hard KL constraint, as in TRPO, can be difficult to optimize in practice because of complex second-order optimization and sensitivity to the constraint threshold (Schulman et al., 2017). To address this, PPO (Schulman et al., 2017) modifies the surrogate objective by clipping the importance sampling ratio, formalized as:

$$L^{\text{CLIP}}(\theta) = \hat{\mathbb{E}}_{(s_t,a_t)\sim\pi_{\theta_{\text{old}}}} \left[\min\left(r_t(\theta)\hat{A}_t,\ \text{clip}(r_t(\theta), 1-\epsilon, 1+\epsilon)\hat{A}_t\right)\right]. \quad (5)$$

This clipping mechanism effectively defines a soft trust region $r_t(\theta)$, which is restricted to remain within a small neighborhood of 1 (controlled by $\epsilon$) to prevent destructive updates. Compared to TRPO, PPO achieves greater stability and efficiency.

## 3 PROXIMAL SUPERVISED FINE-TUNING

PSFT aims to improve performance on the target task while preserving general capabilities beyond the target domain—such as scientific reasoning and instruction following—and preventing entropy collapse (the over-concentration of the output distribution that reduces diversity and harms generalization (Ziegler et al., 2019)), thereby enabling further optimization in the RL stage.

**PSFT.** We revisit the TRPO and PPO objectives in RL. Both methods rely on importance sampling ratios $r_t(\theta)$ to reweight returns, while constraining these ratios to prevent the new policy $\pi_\theta$ from deviating excessively from the old policy $\pi_{\theta_{\text{old}}}$. Translating this idea to the supervised setting with the offline dataset $\mathcal{D}$, where all actions are assumed to be "correct" (i.e., $\hat{A}_t > 0$), we simplify the advantage to $\hat{A}_t = 1$ (as in SFT), and define our PSFT loss as follows:

$$L^{\text{PSFT}}(\theta) = \mathbb{E}_{(s_t,a_t)\sim\mathcal{D}} \left[\min\left(\frac{\pi_\theta(a_t \mid s_t)}{\pi_{\theta_{\text{old}}}(a_t \mid s_t)}, \text{clip}(\frac{\pi_\theta(a_t \mid s_t)}{\pi_{\theta_{\text{old}}}(a_t \mid s_t)}, 1-\epsilon, 1+\epsilon)\right)\right]. \quad (6)$$

This objective regularizes supervised learning updates by limiting the ratio between the new and old policy probabilities. Intuitively, it discourages overconfident changes in token probabilities, thereby optimizing target tasks while preserving existing general capabilities.

**Comparison with PPO.** Although PSFT adopts the soft trust region mechanism used in PPO, it is not a policy-gradient RL method. PPO optimizes expected return using on-policy trajectories and advantage estimates; by contrast, PSFT is a supervised offline objective (here we simplify to $\hat{A}_t = 1$), where the clipped ratio functions mainly as a regularizer that limits overly large changes in token probabilities and reweights gradients.

**Dynamic update of $\pi_{\theta_{\text{old}}}$.** Allowing the old policy $\pi_{\theta_{\text{old}}}$ to evolve dynamically—rather than keeping the initial model $\pi_{\theta_{\text{ref}}}$ fixed—is essential for gradual and stable learning from the dataset. In contrast, fixing the initial model confines optimization to the trust region centered on the reference model, which limits the knowledge that can be effectively leveraged from the offline dataset (see Section 5.3).

**Warm-Up.** $(s_t, a_t)$ in Eq. 6 is sampled from the offline dataset $\mathcal{D}$. In the initial steps, $r_t(\theta)$ may yield a biased expectation since $\pi_{\theta_{\text{old}}}$ may not be aligned with the distribution of $\mathcal{D}$. We let $\pi_{\theta_{\text{old}}}$ evolve during SFT, gradually learning from $\mathcal{D}$, which could largely alleviate the potential bias issue. Besides, a warm-up SFT phase on $\mathcal{D}$ can be optionally introduced to better align the initial old policy $\pi_{\theta_{\text{old}}}$ with the offline dataset, further improving PSFT's target-domain performance (see Section 4.1).

**Gradient Analysis.** Similar to PPO, the gradient is confined to the trust region for fine-tuning:

$$\nabla_\theta L^{\text{PSFT}}(\theta) = \mathbb{E}_{(s_t,a_t)\sim\mathcal{D}} \left[r_t(\theta) \cdot \mathbb{I}_{\text{trust}}(r_t(\theta)) \cdot \nabla_\theta \log \pi_\theta(a_t \mid s_t)\right],$$

$$\text{where } \mathbb{I}_{\text{trust}}(r_t(\theta)) = \begin{cases} 0 & r_t(\theta) > 1+\epsilon, \\ 1 & \text{otherwise.} \end{cases} \quad (7)$$

It shows that if the offline dataset distribution deviates significantly from the model distribution (which is more likely to disturb the general ability), these tokens will have no gradient. This mechanism is **simple yet effective**: it prevents large policy updates, mitigates entropy collapse, and preserves generalization, thereby enabling sustainable progress in the subsequent RL stage. Typically,

the optimal value of $\epsilon$ is set to 0.2 or 0.28 (a larger $\epsilon$ can lead to large gradients). The clipped token cloud is given in Section 5.1, and further analysis is provided in Section 5.2.

## 4 EXPERIMENTS

In Section 4.1, we explore the training dynamics and then evaluate both in-domain and out-of-domain performances. In Section 4.2, we verify the effectiveness of PSFT as the foundation of RL. In Section 4.3, we further conduct experiments in the alignment domain to assess the universal applicability of our approach. In Section 4.4, we further extend PSFT to vision language models.

### 4.1 MAIN EXPERIMENTS ON MATH REASONING IN THE SFT STAGE

**Setup.** *(1) Models and Datasets.* We evaluate our method based on Qwen2.5-7B-Instruct (Yang et al., 2025) and Llama3.1-8B-Instruct (Dubey et al., 2024). Our training data focuses on the math domain, aiming to improve general LLM capabilities through math reasoning. We employ the OpenR1-Math-8192 long chain-of-thought (CoT) dataset (Face, 2025). *(2) Baseline.* We consider the standard SFT method and an SFT variant, denoted as SFT$_{KL}$, which incorporates KL divergence constraints into the loss function. The KL regularization coefficient is set to 0.5, while the $\epsilon$ parameter in PSFT is fixed at 0.28. More details can be found in Appendix C.7.1.

#### 4.1.1 TRAINING DYNAMICS ON SFT

We begin by examining the training dynamics of each method. Specifically, we train each for around 10 epochs and plot the corresponding changes in entropy and performance. We report in-domain performance with AIME-24 avg@32 and out-of-domain performance with GPQA avg@8.

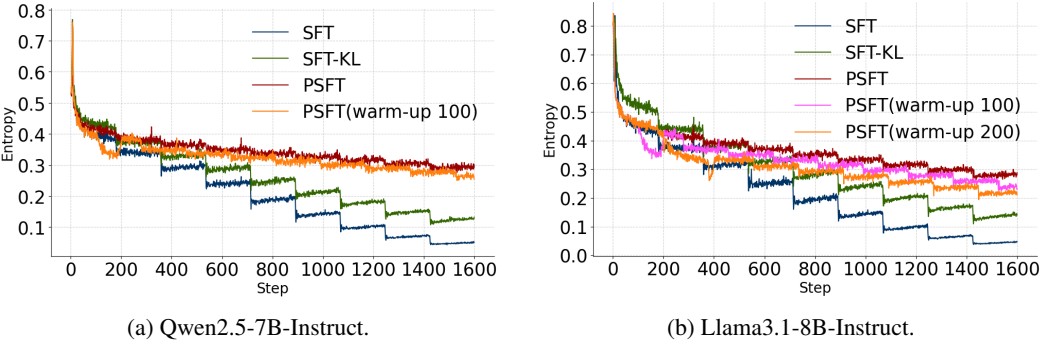

(a) Qwen2.5-7B-Instruct.                    (b) Llama3.1-8B-Instruct.

Figure 1: Training dynamics of entropy. Each epoch contains 178 steps.

**Finding 1: PSFT avoids entropy collapse.** The entropy evolution is shown in Figure 1. Compared to SFT and SFT-KL, our PSFT produces a smoother entropy curve. For SFT and SFT-KL, entropy exhibits a marked decline after each epoch, indicating potential overfitting. This suggests that PSFT is capable of sustaining long-term fine-grained training without triggering entropy collapse. Notably, PSFT with a warm-up phase demonstrates the same stability, and the extent of warm-up plays a critical role in shaping the overall entropy level.

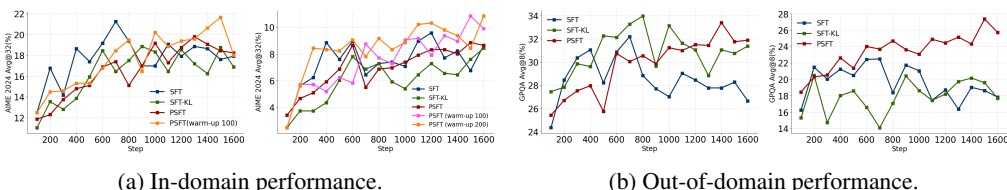

(a) In-domain performance.                  (b) Out-of-domain performance.

Figure 2: Training dynamics of in-domain/out-of-domain performance in SFT experiments. In each subfigure, Qwen2.5-7B-Instruct is shown on the left, and Llama3.1-8B-Instruct on the right.

**Finding 2: PSFT matches or surpasses in-domain performance of standard SFT at a similar entropy level.** Figure 2a illustrates the evolution of in-domain evaluation performance. PSFT

Table 1: Detailed results of the SFT stage. For AIME and AMC, the results are avg.@32. For the other in-domain benchmarks, the results are avg.@8. For GPQA, ARC-C, TruthfulQA, and IFEval, the results are avg.@8. For the remaining out-of-domain benchmarks, the results are pass.@1.

| | Benchmark | Qwen2.5-7B-Instruct | | | | | Llama3.1-8B-Instruct | | | | |
| | | Base | SFT | SFT-KL | PSFT$_{warm-up}$ | PSFT | Base | SFT | SFT-KL | PSFT$_{warm-up}$ | PSFT |
|---|---|---|---|---|---|---|---|---|---|---|---|
| In-Domain | AIME-24 | 11.25 | 22.08 | 19.27 | **22.92** | 19.38 | 3.96 | 10.63 | 9.58 | **12.08** | 10.31 |
| | AIME-25 | 8.75 | 23.02 | 21.56 | **23.02** | 21.98 | 0.73 | 16.77 | 16.15 | **18.75** | 14.48 |
| | AMC | 52.58 | 62.73 | **63.20** | 62.42 | 62.34 | 24.45 | 47.19 | 45.55 | **49.45** | 46.80 |
| | MATH-500 | 75.05 | 84.10 | 83.55 | **84.68** | 83.35 | 48.55 | 72.60 | 69.83 | **74.15** | 71.98 |
| | OlympidBench | 39.72 | 52.35 | **52.70** | 52.30 | 51.50 | 17.09 | 41.43 | 39.19 | **42.07** | 39.61 |
| | Minerva | 40.53 | 43.66 | 42.19 | **43.38** | 43.33 | 25.87 | 32.17 | 26.75 | **33.64** | 32.40 |
| | Avg. | 37.98 | 47.99 | 47.08 | **48.17** | 46.98 | 20.11 | 36.80 | 34.51 | **38.36** | 35.93 |
| Out-of-Domain | GPQA | 31.38 | 32.89 | 32.95 | **33.27** | 33.21 | 24.62 | 19.38 | 18.18 | 23.99 | **26.89** |
| | ARC-C | 91.54 | 92.22 | 91.86 | 92.09 | **92.29** | 80.96 | 87.73 | 87.02 | **89.72** | 89.11 |
| | TruthfulQA | 66.10 | 63.14 | 61.31 | 66.37 | **67.16** | 55.14 | 67.08 | 67.03 | 68.55 | **68.69** |
| | MMLU-Pro | 54.99 | 58.98 | 58.35 | **59.28** | 59.18 | 43.70 | 50.29 | 47.02 | 56.03 | **56.58** |
| | SuperGPQA | 27.59 | 29.02 | 27.69 | **28.37** | 28.10 | 18.16 | 21.95 | 19.81 | 25.26 | **25.85** |
| | HeadQA | 73.41 | 74.65 | 74.07 | 75.24 | **75.82** | 68.20 | **73.52** | 71.95 | 77.71 | **77.90** |
| | IFEval$_{loose}$ | **73.94** | 54.42 | 55.44 | 55.07 | 73.03 | 78.10 | 33.45 | 34.19 | 33.08 | 69.75 |
| | Avg. | 59.85 | 57.90 | 57.38 | 58.53 | **61.26** | 52.70 | 50.49 | 49.31 | 53.48 | **59.25** |

achieves performance on par with, and sometimes exceeding, SFT under comparable entropy levels. For example, at step 1300, PSFT reaches an AIME24 score of nearly 20 with an entropy of around 0.3. This entropy corresponds to that of SFT between epochs 300 and 520, yet PSFT delivers a higher AIME score within this range. SFT with KL remains less effective than PSFT when considering both entropy dynamics and in-domain performance.

**Finding 3: PSFT with warm-up surpasses standard SFT.** Figure 2a shows the effect of adding a warm-up phase to PSFT. The in-domain performance improves steadily, with PSFT + warm-up consistently outperforming both vanilla PSFT and standard SFT baselines. Furthermore, longer warm-up periods yield even greater gains in in-domain performance.

**Finding 4: PSFT avoids large fluctuations in policy updates.** As observed in both in-domain and out-of-domain evaluations (GPQA, Figure 2b), PSFT generally maintains an upward performance trend. In contrast, SFT and SFT-KL exhibit pronounced fluctuations in out-of-domain performance. Therefore, PSFT preserves the better generalization to out-of-domain tasks.

### 4.1.2 DETAILED EVALUATIONS

**Setup.** *(1) Model setup:* We select checkpoints by in-domain performance: for Qwen2.5-7B-Instruct, the SFT model at 700 steps, SFT-KL at 900 steps, and PSFT at 1300 steps. *(2) Evaluation benchmark*: See Appendix C.7.1. *(3) Inference.* The inference length is set to 10,240 tokens (4,096 for IFEval), with a top-$p$ of 0.95 and a temperature of 0.7.

**Finding 1: PSFT with warm-up consistently outperforms standard SFT on in-domain tasks.** Table 1 presents the performance across various in-domain benchmarks, where all fine-tuned methods achieve notable improvements over the Base model. PSFT with the warm-up phase achieves the best overall results, maintaining robustness across different in-domain benchmarks.

**Finding 2: PSFT demonstrates strong generalization ability.** Table 1 also shows different performance on out-of-domain benchmarks. We find that injecting long CoT data into the model improves its general reasoning ability, consistent with the findings reported in Zhou et al. (2025). In contrast, PSFT exhibits a significantly better generalization ability compared to SFT. For example, in IFEval, which evaluates instruction-following capability, SFT, SFT-KL, and PSFT with warm-up significantly degrade the original results. In contrast, PSFT not only maintains strong in-domain performance but also significantly improves reasoning abilities on diverse out-of-domain tasks, with only minimal compromise in instruction compliance.

**Finding 3: PSFT demonstrates robustness across models.** For example, when evaluating GPQA and TruthfulQA tasks, SFT and SFT-KL exhibit varying behaviors across different base models—one sometimes compromises performance while the other may boost it. In contrast, PSFT consistently improves performance on both Qwen and Llama models.

Table 2: Detailed results of the RL stage. For AIME and AMC, the results are avg.@32. For the other in-domain benchmarks, the results are avg.@8. For GPQA, ARC-C, TruthfulQA, and IFEval, the results are avg.@8. For the remaining out-of-domain benchmarks, the results are pass.@1.

| | Benchmark | Qwen2.5-7B-Instruct | | | | Llama3.1-8B-Instruct | | | |
| | | SFT | SFT → GRPO | PSFT | PSFT → GRPO | SFT | SFT → GRPO | PSFT | PSFT → GRPO |
|---|---|---|---|---|---|---|---|---|---|
| In-Domain | AIME-24 | 22.08 | 27.40 | 19.38 | **28.13** | 10.63 | 14.06 | 10.31 | **14.27** |
| | AIME-25 | 23.02 | 26.15 | 21.98 | **27.19** | 16.77 | 17.50 | 14.48 | **18.02** |
| | AMC | 62.73 | 70.86 | 62.34 | **71.72** | 47.19 | 54.38 | 46.80 | **55.23** |
| | MATH-500 | 84.10 | 86.60 | 83.35 | **87.48** | 72.60 | 77.03 | 71.98 | **77.78** |
| | OlympidBench | 52.35 | 58.09 | 51.50 | **58.50** | 41.43 | 47.13 | 39.61 | **47.74** |
| | Minerva | 43.66 | 45.27 | 43.33 | **46.83** | 32.17 | 33.09 | 32.40 | **37.55** |
| | Avg. | 47.99 | 52.40 | 46.98 | **53.31** | 36.80 | 40.53 | 35.93 | **41.77** |
| Out-of-Domain | GPQA | 32.89 | 39.39 | 33.21 | **43.43** | 19.38 | 31.06 | 26.89 | **36.23** |
| | ARC-C | 92.22 | 92.25 | 92.29 | **92.42** | 87.73 | 88.57 | 89.11 | **90.92** |
| | TruthfulQA | 63.14 | 61.93 | **67.16** | 64.84 | 67.08 | 64.82 | **68.69** | 65.24 |
| | MMLU-Pro | 58.98 | 62.61 | 59.18 | **63.65** | 50.29 | 54.39 | 56.58 | **60.92** |
| | SuperGPQA | 29.02 | 33.80 | 28.10 | **34.51** | 21.95 | 27.80 | 25.85 | **31.30** |
| | HeadQA | 74.65 | 75.42 | 75.82 | **75.89** | 73.52 | 73.81 | 77.90 | **78.12** |
| | IFEval$_{loose}$ | 54.42 | 53.89 | 73.03 | **73.73** | 33.45 | 31.85 | 69.75 | **71.02** |
| | Avg. | 57.90 | 59.90 | 61.26 | **64.06** | 50.49 | 53.19 | 59.25 | **61.96** |

## 4.2 EXPLORATION ON THE POTENTIAL OF MODELS IN THE RL STAGE

Practical LLMs often conduct RL after SFT in the post-training phase. The SFT models should serve as an appropriate starting point for the subsequent RL stage, avoiding both overfitting and underfitting, and thereby better stimulate the potential of RL. We evaluate the power of PSFT in the RL stage.

**Setup.** We adopt DAPO (Yu et al., 2025) with a clip-higher value of 0.28, a stable variant of GPPO. The RL training uses the DAPO-MATH-17k dataset, with detailed training configurations provided in Appendix C.7.2. We select the checkpoints based on the highest in-domain performance for evaluation.

### 4.2.1 TRAINING DYNAMICS ON RL

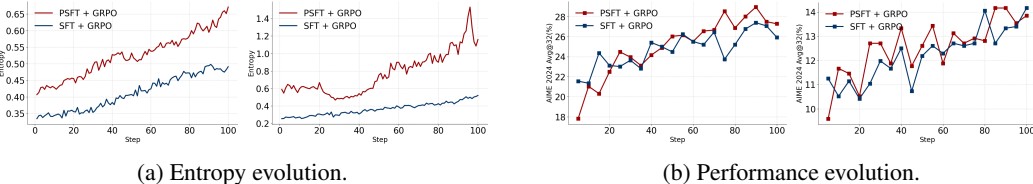

(a) Entropy evolution.  (b) Performance evolution.

Figure 3: Training dynamics of entropy and performance in RL experiments. In each subfigure, Qwen2.5-7B-Instruct is shown on the left, and Llama3.1-8B-Instruct on the right.

**Finding 1: PSFT leaves much room for RL optimization.** The RL entropy evolution is shown in Figure 3a. As low entropy limits the model's exploration, it is notable that using PSFT as a cold start results in higher entropy throughout training, with a steeper increase.

**Finding 2: PSFT shows slow start and rapid catch-up in RL.** The in-domain performance evolution is presented in Figure 3b. Thanks to its high entropy, which promotes exploration, using PSFT as the cold start leads to a relatively steep performance improvement and ultimately surpasses the performance achieved when using the standard SFT as the cold start.

### 4.2.2 DETAILED EVALUATION

**Finding 3: PSFT achieves better in-domain performance after RL.** Table 2 presents the results of RL experiments on in-domain benchmarks. Overall, RL improves the performance of all models. A comprehensive evaluation of six in-domain tasks shows that, although PSFT initially lags behind SFT, it holds significant potential that can be further unlocked through RL.

**Finding 4: PSFT still outperforms the standard SFT on out-of-domain tasks by a large margin after RL.** Table 2 also presents the results of RL experiments on out-of-domain benchmarks. It is evident that if the model exhibits weak capabilities during the cold-start phase, the subsequent RL training is constrained by these limitations (e.g., the results of IFEval). PSFT functions well on different base models. It further underscores the necessity of enhancing the current SFT algorithm.

## 4.3 FURTHER EXPERIMENTS ON HUMAN ALIGNMENT

In this section, we demonstrate the universality of PSFT by extending our experiments to human alignment datasets, employing various base models and algorithms, such as DPO (Rafailov et al., 2023).

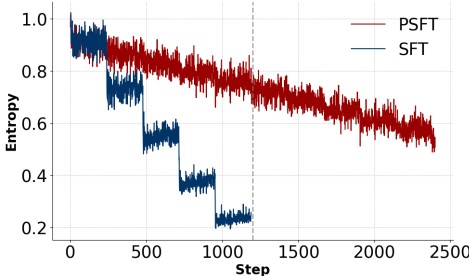 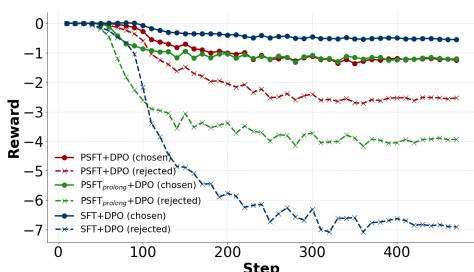

(a) Entropy evolution during the SFT stage.

(b) Positive/Negative reward progression during DPO training.

Figure 4: Training dynamics of human alignment experiments.

Our results show that PSFT effectively reduces the alignment tax (i.e., the generalization gap) while still leaving room for further optimization.

**Setup.** We adopt a completely different setup to demonstrate the universality of PSFT further. *(1) Model and Dataset.* We fine-tune the pre-trained Qwen3-4B-Base (Yang et al., 2025) model with the UltraFeedback dataset (Cui et al., 2023). *(2) Training pipeline.* We select the chosen region of the dataset and then perform SFT/PSFT. And for PSFT, we additionally train the double steps named PSFT$_{\text{prolong}}$. Then, use the DPO algorithm to enable the model to learn the contrastive reward signal. *(3) Evaluation.* See Appendix C.7.3.

**Finding 1: PSFT reliably prevents entropy collapse and enables DPO for alignment.** As shown in Figure 4a, the entropy loss of SFT still shows a severe sawtooth shape, indicating a potential overfitting phenomenon, while PSFT overcomes this issue. Therefore, SFT leads to a severe alignment tax (Figure 5) and restricts further optimization. Table 3 presents the results of DPO experiments on alignment benchmarks. We observe that PSFT$_{\text{prolong}}$ achieves a similar effect to SFT while avoiding entropy collapse, and it performs even better during DPO training. PSFT provides a strong starting point, with DPO training unlocking further gains.

**Finding 2: PSFT training exploits both positive and negative samples well.** As shown in Figure 4b, the cold-start point can influence the subsequent DPO phase. PSFT$_{\text{prolong}}$ and PSFT yield similar rewards on positive samples; the main difference is that prolonged training reduces the occurrence of negative samples. In contrast, SFT training results in positive samples appearing frequently while negative samples are rare, which limits the

Table 3: Qwen3-4B-Base DPO training on the alignment benchmarks.

| Method | AlpacaEval2 | | Arena-Hard | MT-Bench | |
|---|---|---|---|---|---|
| | LC (%) | WR (%) | WR (%) | 1-turn | 2-turn |
| SFT | 12.24 | 8.52 | 17.90 | 7.64 | 5.71 |
| ↪ DPO | 16.96 | 13.40 | 26.50 | 7.91 | 6.00 |
| PSFT$_{\text{prolong}}$ | 11.95 | 8.37 | 17.40 | 7.41 | 5.84 |
| ↪ DPO | 19.26 | 15.17 | 30.20 | 7.63 | 6.74 |
| PSFT | 11.79 | 7.49 | 11.40 | 6.83 | 4.86 |
| ↪ DPO | **23.29** | **20.13** | **36.40** | **8.51** | **6.95** |

model's ability to learn from the reward signal and instead biases it toward the human-selected data. This is consistent as reported by Xiao (2024) that when an SFT model becomes overly biased toward certain outputs with near certainty, it can subsequently induce preference collapse in the alignment process.

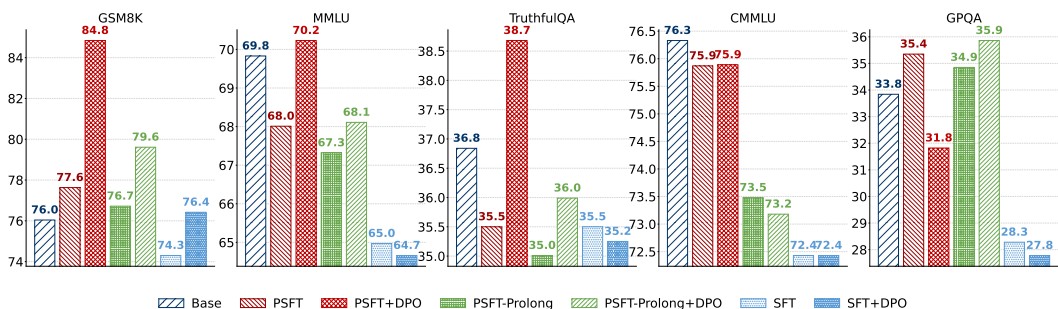

Figure 5: Results on alignment tax (out-of-domain tasks).

Table 4: Detailed results of in-domain text-only evaluation.

| Benchmark | Base | SFT | PSFT |
|---|---|---|---|
| AIME-24 | 6.15 | 8.54 | **9.27** |
| AIME-25 | 1.88 | 10.21 | **10.73** |
| AMC | 34.77 | 44.69 | **44.84** |
| MATH-500 | 65.65 | **71.70** | 70.58 |
| OlympidBench | 29.44 | **37.50** | 36.02 |
| Minerva | 30.93 | 31.85 | **32.26** |
| MMLU | **63.10** | 45.39 | 62.44 |
| MMLU-Pro | 47.90 | 40.67 | **50.55** |

Table 5: Detailed results of multi-modal evaluation.

| Benchmark | Geo3K | MathVerse | MATHVista | MMMU | MMMU-Pro |
|---|---|---|---|---|---|
| Base | 29.28 | 37.03 | 66.80 | 41.89 | **29.59** |
| ↪ GRPO | 49.42 | 42.44 | 71.10 | 40.00 | 28.61 |
| SFT | 33.78 | 38.83 | 68.00 | 40.78 | 23.58 |
| ↪ GRPO | 48.92 | 43.12 | **72.70** | 38.78 | 19.60 |
| PSFT | 33.61 | 38.25 | 69.60 | **43.33** | 27.63 |
| ↪ GRPO | **50.58** | **44.14** | 72.20 | 40.78 | 26.07 |

**Finding 3: PSFT reduces the alignment tax.** Figure 5 shows the alignment tax results with various tasks (e.g., GSM8K, MMLU, GPQA) as the out-of-domain tasks. It indicates that: (a) PSFT and PSFT+DPO could largely maintain the general ability during the SFT stage compared to the standard SFT. (b) PSFT-Prolong, even being degraded by possible overfitting issues, achieves relatively good results on alignment tax, implying the robustness of PSFT in practical scenarios.

## 4.4 FURTHER EXPLORATIONS OF PSFT ON VISION LANGUAGE MODELS

**Setup.** *(1) Model and Datasets.* We employ the Qwen2.5-7B-VL-Instruct (Bai et al., 2025) vision language model (VLM). We utilize the text-only OpenR1-Math-4096 dataset (Face, 2025) and the Geometry-3k (Lu et al., 2021) dataset, which consists of text-image geometry problems. *(2) Training Pipeline.* We conduct SFT/PSFT on the VLM using the text-only dataset to inject the long CoT reasoning ability. After that, we perform RL training on the text-image pair dataset to further enhance the visual reasoning ability. *(3) Evaluation.* Motivation and additional experimental details are provided in Appendix C.7.4.

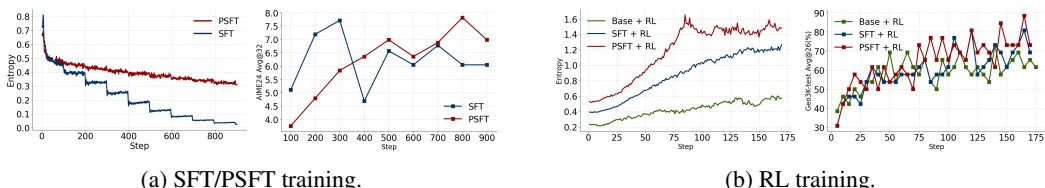

(a) SFT/PSFT training.                    (b) RL training.

Figure 6: Training dynamics of Entropy and performance on VLM experiments.

**Finding 1: PSFT functions well on multimodal scenarios.** Based on Figure 6, PSFT still effectively avoids entropy collapse while achieving comparable in-domain performance. By maintaining sufficient entropy space, the model maintains excellent performance throughout the RL process. It suggests that incorporating text-only long CoT data can enhance the reasoning ability of VLMs.

**Finding 2: PSFT preserves the general ability, while SFT compromises.** Table 4 shows that PSFT and SFT successfully inject the long reasoning data into the VLM models, achieving great improvement on in-domain text-only benchmarks. However, MMLU and MMLU-Pro indicate that

SFT greatly weakens performance in general domains. Table 5 shows similar results. In-domain text-only long CoT data can improve a VLM's performance on in-domain multimodal reasoning. However, results on MMMU and MMMU-Pro indicate that SFT suffers a significant drop in multi-disciplinary multimodal understanding. In contrast, PSFT consistently maintains strong performance, achieving the best overall results across both in-domain and out-of-domain benchmarks.

## 5 IN-DEPTH ANALYSES

### 5.1 CLIPPED TOKENS IN PSFT

Figure 7 presents representative clipped tokens observed during the PSFT training process in Exp. 4.1. The clipped token mostly concentrates on uncertain words like "wait", "alternatively", and similar words that reflect certain "long thinking patterns". With the training in progress, these token-clip weights are more pronounced, while other tokens become smaller. The "thinking pattern" has been gradually and smoothly incorporated into models via PSFT with minimal disturbance to the general capabilities.

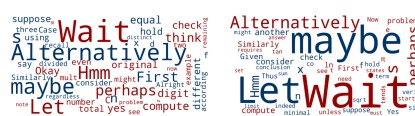

(a) Clipped token cloud in epoch 1. (b) Clipped token cloud in epoch 3.

Figure 7: Evolution of clipped tokens.

### 5.2 PARAMETER ANALYSIS ON CLIPPED VALUE

As shown in Figure 8, PSFT without clipping maintains higher entropy but suffers from large, unstable gradient norms and highly fluctuating downstream results. By introducing the trust-region clipping mechanism, PSFT strikes a better balance: it prevents entropy collapse, stabilizes gradient updates, and achieves steady improvements in performance. Across different clipping thresholds, with medium values of $\epsilon$ (e.g., 0.28) showing particularly favorable trade-offs between stability and performance. Flexible choices can be made within an appropriate range of $\epsilon$ for practical demands.

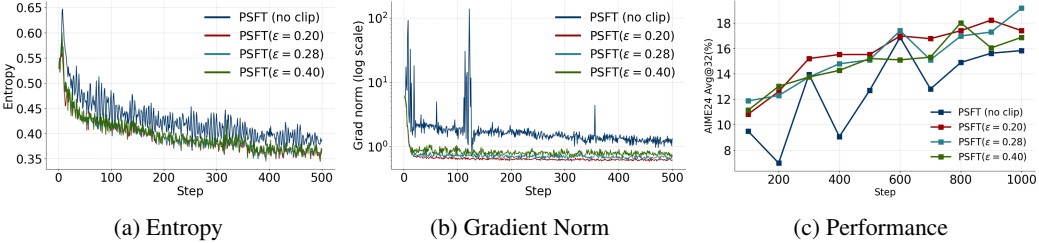

(a) Entropy      (b) Gradient Norm      (c) Performance

Figure 8: Training dynamics and in-domain results of PSFT with different clipped values.

### 5.3 UPDATE FREQUENCY OF THE OLD POLICY

Figure 9 and Table 6 show that dynamically updating the old policy $\pi_{\theta_{\text{old}}}$ is essential for effective PSFT. While fixed references (no updates) still yield improvements over the base model, their overall performance remains poor. In contrast, frequent updates—especially every 4 steps—deliver substantial gains on in-domain tasks. For update intervals of

Table 6: Results with different frequencies.

| Benchmark | AIME-24 | AIME-25 | TruthfulQA | IFEval$_{\text{loose}}$ |
|---|---|---|---|---|
| Base | 11.25 | 8.75 | 66.10 | 73.94 |
| PSFT (no upd) | 13.02 | 11.15 | 66.61 | **76.46** |
| PSFT (upd.16) | 21.56 | **23.54** | 66.10 | 69.33 |
| PSFT (upd.8) | 19.38 | 21.98 | **67.16** | 73.03 |
| PSFT (upd.4) | **22.50** | 22.92 | 66.72 | 72.08 |
| SFT | 22.08 | 23.02 | 63.14 | 54.42 |

16 steps (batch size 256, mini-batch size 16) and 8 steps (mini-batch size 32), we use the same learning rate of $1e - 6$. These results further indicate that smaller batch sizes, combined with relatively higher learning rates, can enhance in-domain performance, albeit at the expense of generalization due to high fluctuations.

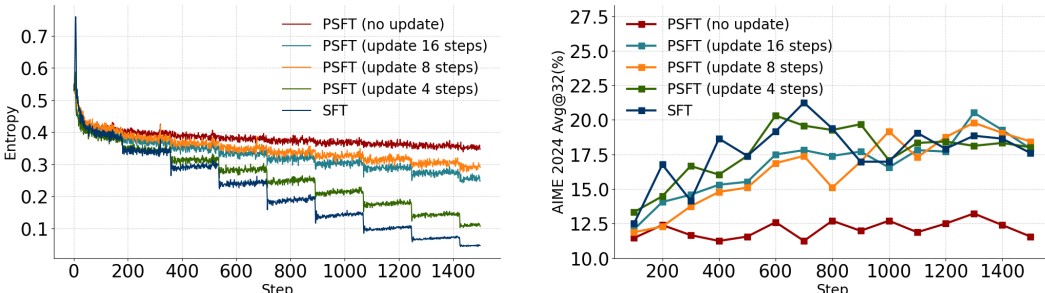

Figure 9: Training Dynamics under different old policy model updating frequencies.

## 6 RELATED WORK

**Supervised Fine-tuning.** SFT, typically as the first post-training step, sets the stage for a more robust fine-tuned model. Several studies have noted that the standard cross-entropy (CE) loss may not be the most suitable objective for SFT (Li et al., 2024b; Xiao, 2024), arguing that training with CE at this stage tends to encourage the model to memorize the training data rather than acquire more generalizable capabilities. One line of research focuses on improving target performance by addressing the distribution mismatch between expert demonstrations and the evolving target policy. iw-SFT (Qin & Springenberg, 2025) reinterpret standard SFT as optimizing a loose lower bound on the RL objective. This lower bound becomes increasingly loose as the model distribution drifts away from the reference policy. iw-SFT addresses this gap by introducing importance re-weighting, which assigns higher weights to more preferred trajectories and thus tightens the bound toward the true RL objective. DFT (Wu et al., 2025) views SFT as a flawed policy gradient and introduces a simple probability-based reweighting that improves targeted performance. **Our work aims to prevent entropy collapse while enhancing target performance, without compromising general capability, and leaving room for further optimization.**

**Reinforcement Learning.** Following SFT, RL is employed to optimize reward signals directly. Policy gradient methods (Sutton et al., 2000) provide the foundation but are prone to instability and high variance. To address this, TRPO (Schulman et al., 2015) introduces a trust-region constraint, limiting each policy update to a small KL-divergence neighborhood to ensure stable improvement. PPO (Schulman et al., 2017) further simplifies this approach with a clipped surrogate objective, striking a balance between stability and efficiency. GRPO (Shao et al., 2024) removes the critic and computes advantage using the relative rewards of outputs within each group, treating all tokens in a sequence equally. **We extend the benefits of these methods to the supervised setting.**

**Combination of SFT and RL.** Recent studies have also investigated hybrid objectives that couple SFT loss on offline data with RL loss on online rollouts (Kang et al., 2023; Yan et al., 2025). LUFFY (Yan et al., 2025) exemplifies this direction by mixing offline demonstrations with online trajectories at a fixed proportion in each training batch. Extending this approach, Hybrid Post-Training (HPT) (Lv et al., 2025) adaptively switches between SFT and RL based on rollout performance, balancing guidance and exploration to enhance reasoning. This idea of constructing a composite loss has also been adopted in several recent frameworks (Zhang et al., 2025; Liu et al., 2025). **Our method is orthogonal to these methods.**

## 7 CONCLUSION AND FUTURE WORK

Motivated by TRPO and PPO, we propose Proximal Supervised Fine-Tuning (PSFT). PSFT preserves a smooth entropy curve while achieving performance comparable to standard SFT, and it exhibits strong generalization. Furthermore, using PSFT as a cold-start model facilitates more effective post-training techniques such as RL optimization and DPO. Overall, PSFT presents a promising alternative to traditional SFT. In the future, we will explore the universality of our PSFT on more diverse, industry-level datasets and models.

## REPRODUCIBILITY STATEMENT

Our work is easy to reproduce, as mainstream RL frameworks can support this algorithm with minimal code modifications. You can replace rollout generation with SFT demonstrations and assign a constant value to all advantages in the sequence for training. We provide our code, built upon the open-source `verl` framework, in the supplementary material. Detailed hyperparameter settings are provided in the Appendix.

## ETHICS STATEMENT

All datasets and models used in this study for our experiments are publicly available and open-source.

## ACKNOWLEDGEMENT

This work is supported by the AI for Science Program, Shanghai Municipal Commission of Economy and Informatization AI (2025-GZL-RGZN-BTBX-01013), and the Young Elite Scientists Sponsorship Program by CAST (2023QNRC001).

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

# A  LIMITATION

Our experiments are currently limited in terms of model scale, dataset size, and data diversity. Future work is needed to evaluate the effectiveness of PSFT on industrial-scale models, larger datasets, and mixed data. We have not yet tested our approach on the latest model architectures, such as MoE or hybrid models, which will be considered in future studies.

# B  LLM-USAGE STATEMENT

We use LLMs to polish writing and correct grammar.

# C  APPENDIX

## C.1  CONVERGENCE ANALYSIS

**Assumption 1 (Smoothness of the NLL)** *The negative log likelihood is $L$–smooth, i.e.*
$$\left\|\nabla_\theta^2 \log \pi_\theta(a \mid s)\right\| \leq L, \qquad \forall (s, a) \in \mathcal{S} \times \mathcal{A}.$$

**Assumption 2 (Bounded stochastic gradients)** *There exists $G > 0$ such that*
$$\left\|\nabla_\theta \log \pi_\theta(a \mid s)\right\| \leq G, \qquad \forall (s, a) \in \mathcal{S} \times \mathcal{A}.$$

**Theorem 1 (Convergence Rate )** *Under Assumptions 1 and 2, running gradient ascent with learning rate $\alpha_t = \frac{\alpha}{\sqrt{t}}$ for $T$ iterations yields:*

$$\frac{1}{T}\sum_{t=1}^{T}\mathbb{E}\left\|\nabla L^{\mathrm{PSFT}}(\theta_t)\right\|^2 \;\leq\; \frac{2\big(L^{\mathrm{PSFT}}(\theta^*) - L^{\mathrm{PSFT}}(\theta_1)\big)}{\alpha\sqrt{T}} + \frac{\alpha L G^2 (1+\epsilon)^2 (1+\ln T)}{2T} \tag{8}$$

***Proof.*** Denote by $F(\theta) := L^{\mathrm{PSFT}}(\theta)$ the objective. Let $g_t$ be the (mini-batch) stochastic gradient at iteration $t$:
$$g_t \;=\; \big[\, r_t(\theta)\, \mathbb{I}_{\mathrm{trust}}\big(r_t(\theta)\big)\, \nabla_\theta \log \pi_\theta(a_t \mid s_t)\big], \quad \text{so that} \quad \mathbb{E}\big[g_t \mid \theta_t\big] = \nabla F(\theta_t).$$
Because $r_t(\theta) \leq 1 + \epsilon$ in the trust region, Assumption 2 implies $\mathbb{E}\|g_t\|^2 \leq (1+\epsilon)^2 G^2 := \bar{G}^2$.

**(1) Smoothness of $F$.**  For any $\theta, \Delta$,
$$F(\theta + \Delta) \;\leq\; F(\theta) + \nabla F(\theta)^\top \Delta + \frac{L_{\mathrm{clip}}}{2}\|\Delta\|^2, \qquad L_{\mathrm{clip}} := L(1+\epsilon).$$
The same inequality with the opposite sign also holds, so $F$ is $L_{\mathrm{clip}}$–smooth.

**(2) One-step progress.**  With the ascent update $\theta_{t+1} = \theta_t + \alpha_t g_t$ and the lower bound of $L_{\mathrm{clip}}$–smoothness,
$$F(\theta_{t+1}) \geq F(\theta_t) + \alpha_t \nabla F(\theta_t)^\top g_t - \frac{L_{\mathrm{clip}}}{2}\alpha_t^2 \|g_t\|^2. \tag{9}$$
Taking conditional expectation $\mathbb{E}_t[\cdot] := \mathbb{E}[\cdot \mid \theta_t]$,
$$\mathbb{E}_t\big[F(\theta_{t+1})\big] \;\geq\; F(\theta_t) + \alpha_t \|\nabla F(\theta_t)\|^2 - \frac{L_{\mathrm{clip}}}{2}\alpha_t^2 \bar{G}^2. \tag{10}$$

**(3) Bounding the gradient norm.**  Rearranging equation 10 yields
$$\alpha_t \|\nabla F(\theta_t)\|^2 \;\leq\; \mathbb{E}_t\big[F(\theta_{t+1})\big] - F(\theta_t) + \frac{L_{\mathrm{clip}}}{2}\alpha_t^2 \bar{G}^2.$$
Summing $t = 1$ to $T$ and taking full expectation,
$$\sum_{t=1}^{T} \alpha_t \,\mathbb{E}\|\nabla F(\theta_t)\|^2 \;\leq\; \mathbb{E}F(\theta_{T+1}) - F(\theta_1) + \frac{L_{\mathrm{clip}}}{2}\bar{G}^2 \sum_{t=1}^{T}\alpha_t^2. \tag{11}$$
Let $\Delta_F := F^* - F(\theta_1)$ where $F^* := \sup_\theta F(\theta)$. Because $F(\theta_{T+1}) \leq F^*$, equation 11 becomes
$$\sum_{t=1}^{T} \alpha_t \,\mathbb{E}\|\nabla F(\theta_t)\|^2 \;\leq\; \Delta_F + \frac{L_{\mathrm{clip}}}{2}\bar{G}^2 \sum_{t=1}^{T}\alpha_t^2.$$

**(4) Choosing the step size.** Choose $\alpha_t = \frac{\alpha}{\sqrt{t}}$ and assume $\alpha \leq 1/L_{\text{clip}}$ (so that $1 - \frac{L_{\text{clip}}\alpha_t}{2} \geq \frac{1}{2}$). Then

$$\sum_{t=1}^{T} \alpha_t \geq \alpha \sum_{t=1}^{T} \frac{1}{\sqrt{t}} \geq \alpha(2\sqrt{T} - 2) \geq \alpha\sqrt{T},$$

$$\sum_{t=1}^{T} \alpha_t^2 = \alpha^2 \sum_{t=1}^{T} \frac{1}{t} \leq \alpha^2(1 + \ln T).$$

**(4) Averaged bound.** Divide equation 11 by $T$ and use the two bounds above:

$$\frac{1}{T} \sum_{t=1}^{T} \mathbb{E}\|\nabla F(\theta_t)\|^2 \leq \frac{\Delta_F}{\alpha\sqrt{T}} + \frac{\alpha L_{\text{clip}}\bar{G}^2(1 + \ln T)}{2T}.$$

Substituting $L_{\text{clip}} = L(1 + \epsilon)$ and $\bar{G} = (1 + \epsilon)G$ gives

$$\frac{1}{T} \sum_{t=1}^{T} \mathbb{E}\|\nabla L^{\text{PSFT}}(\theta_t)\|^2 \leq \frac{2(L^{\text{PSFT}}(\theta^*) - L^{\text{PSFT}}(\theta_1))}{\alpha\sqrt{T}} + \frac{\alpha L G^2(1 + \epsilon)^2(1 + \ln T)}{2T}.$$

When $T \to \infty$ the second term vanishes as $\mathcal{O}(\ln T/T)$, so the dominant rate is $\mathcal{O}(1/\sqrt{T})$. This is consistent with the optimal upper bound of the standard non-convex SGD, indicating that under reasonable assumptions of smoothness and gradient bound, the stochastic gradient rise of PSFT has the same theoretical convergence rate as that of ordinary NLL-SFT.

$\square$

## C.2 Entropy As advantage estimation

We perform training on the high-entropy tokens (Wang et al., 2025).

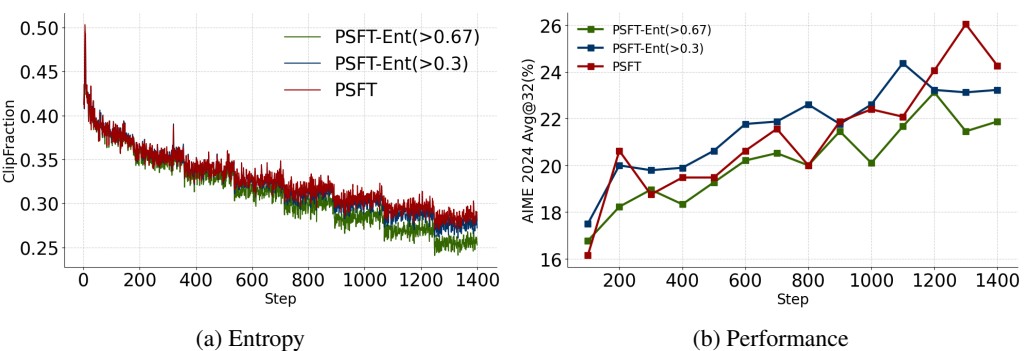

(a) Entropy          (b) Performance

Figure 10: Training dynamics and in-domain results of PSFT with and without entropy token selection.

| Benchmark | AIME-24 | AIME-25 | TruthfulQA |
|---|---|---|---|
| PSFT | **26.98** | **28.75** | 62.90 |
| PSFT-Ent( $> 0.3$ ) | 25.21 | 28.65 | 62.48 |
| PSFT-Ent( $> 0.672$ ) | 24.17 | 25.10 | **63.71** |

Table 7: Results under different baselines.

**Results.** We believe that this sparse update may also contribute to more robust generalization.

## C.3 More large Models

We test PSFT on Qwen3-30B-A3B-2507-Instruct MoE model.

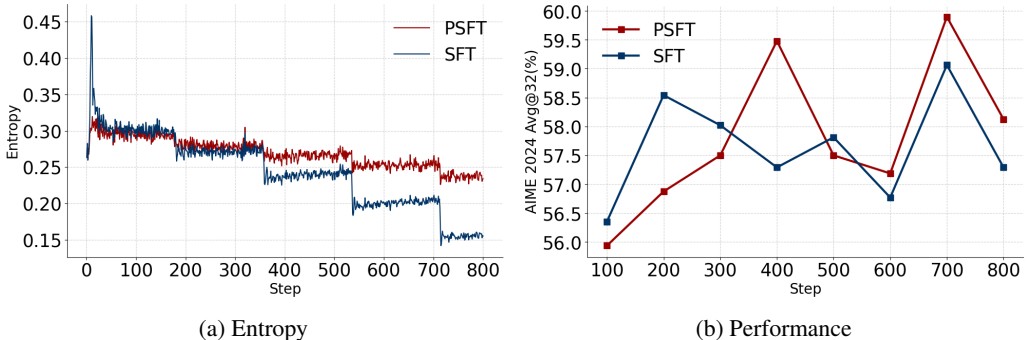

(a) Entropy               (b) Performance

Figure 11: Training dynamics and in-domain results of PSFT with different methods .

| Benchmark | AIME-24 | AIME-25 | TruthfulQA | IFEval$_{\text{loose}}$ |
|---|---|---|---|---|
| PSFT | **61.67** | **48.38** | **80.19** | **88.52** |
| SFT | 59.89 | 46.88 | 77.89 | 50.56 |

Table 8: Results under different baselines.

**Results.** The results show that PSFT serves as a powerful and practical tool for industrial-scale model training.

## C.4 RECENT BASELINES

**Baseline 1: DFT.** DFT (Wu et al., 2025) views SFT as a flawed policy gradient and introduces a simple probability-based reweighting that improves targeted performance.

$$\mathcal{L}_{\text{DFT}}(\theta) = \mathbb{E}_{(s_t,a^\star)\sim\mathcal{D}} \left[ -\sum_{t=1}^{|a^\star|} \text{sg}\left(\pi_\theta\left(a_t^\star \mid s_t\right)\right) \log \pi_\theta\left(a_t^\star \mid s_t\right) \right]. \tag{12}$$

**Baseline 2: Iw-SFT** iw-SFT (Qin & Springenberg, 2025) reframes standard SFT as optimizing a loose lower bound of the RL objective. This bound becomes increasingly loose as the model distribution diverges from the reference policy. To mitigate this mismatch, iw-SFT introduces importance re-weighting, assigning higher weights to more preferred trajectories and thereby tightening the bound toward the true RL objective. In addition, iw-SFT incorporates a clipping range in the importance weights; following the settings in the original paper, we adopt hyperparameter values of 0.2 and 1.8.

$$\mathcal{L}_{\text{iw}-\text{SFT}}(\theta) = \mathbb{E}_{(s_t,a^\star)\sim\mathcal{D}} \left[ \frac{\pi_\theta\left(a_t^\star \mid s_t\right)}{\pi_{\text{ref}}(a_t^* \mid s_t)} \log \pi_\theta\left(a_t^\star \mid s_t\right) \right]. \tag{13}$$

| Benchmark | AIME-24 | AIME-25 | TruthfulQA | IFEval$_{\text{loose}}$ |
|---|---|---|---|---|
| Base | 11.25 | 8.75 | 66.10 | **73.94** |
| PSFT (lr 1e-6, bs 32) | 19.38 | 21.98 | **67.16** | 73.03 |
| PSFT (lr 3e-6, bs 32) | **22.50** | **22.92** | 66.63 | 71.32 |
| iw-SFT | 17.64 | 20.31 | 63.67 | 43.92 |
| DFT | 10.73 | 10.10 | 54.53 | 47.34 |

Table 9: Results under different baselines.

**Results.** As shown in Figure 12, DFT exhibits markedly different training dynamics and yields poor performance, which may indicate that it only benefits from domain-specific signals. In con-

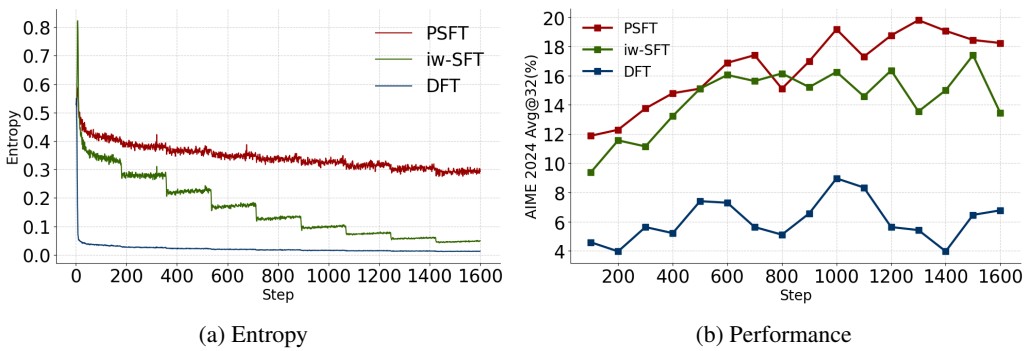

(a) Entropy

(b) Performance

Figure 12: Training dynamics and in-domain results of PSFT with different methods.

trast, iw-SFT shows training behavior similar to PSFT, though PSFT ultimately achieves better performance.

## C.5 LOWER LEARNING RATE

In this section, we examine the impact of the learning rate on both PSFT and SFT. As shown in Figure 14, a large learning rate induces greater update fluctuations, which in turn results in a higher clip fraction. Moreover, increasing the learning rate enables the model to fit the data distribution more rapidly. Based on Table 10, PSFT still achieves stable performance due to the gradient clipping mechanism, and the evaluation on IFEval$_{loose}$ shows that PSFT outperforms SFT with a small learning rate.

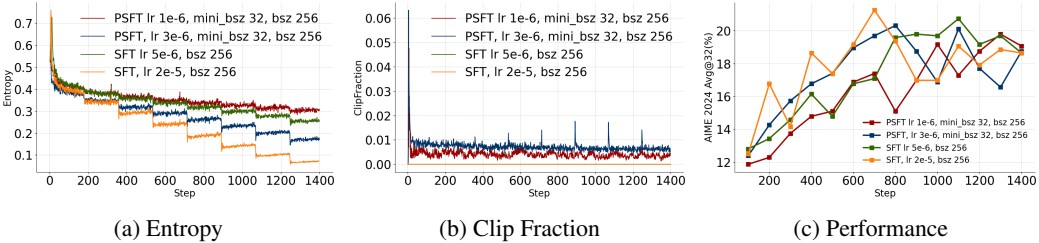

(a) Entropy

(b) Clip Fraction

(c) Performance

Figure 13: Training dynamics and in-domain results of PSFT with different clipped values.

| Benchmark | AIME-24 | AIME-25 | TruthfulQA | IFEval$_{loose}$ |
|---|---|---|---|---|
| Base | 11.25 | 8.75 | 66.10 | **73.94** |
| PSFT (lr 1e-6, bs 32) | 19.38 | 21.98 | **67.16** | 73.03 |
| PSFT (lr 3e-6, bs 32) | **22.50** | **22.92** | 66.63 | 71.32 |
| SFT (lr 2e-5, bs 256) | 22.08 | 23.02 | 63.14 | 54.42 |
| SFT (lr 5e-6, bs 256) | 21.25 | 22.50 | 66.54 | 68.87 |

Table 10: Results under different learning rates across the two methods.

## C.6 CODING TASK

We use the OpenR1-code dataset and filter it for lengths less than 8192, composing the 13k dataset. We use LiveCodeBench as the evaluation (Jain et al., 2025).

**Results.** These results show that while both SFT and PSFT effectively learn the target domain, PSFT better preserves mathematical instruction-following ability.

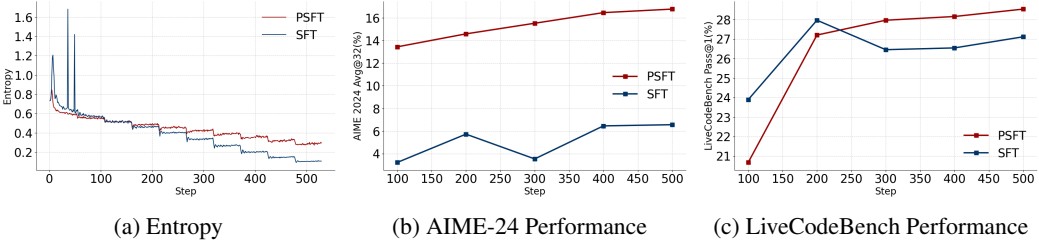

(a) Entropy      (b) AIME-24 Performance      (c) LiveCodeBench Performance

Figure 14: Training dynamics and in-domain results of PSFT with different clipped values.

## C.7 EXPERIMENTAL DETAILS

We perform SFT, PSFT, and RL training using the `verl` framework (Sheng et al., 2024), and employ `LLama-Factory` (Zheng et al., 2024b) for DPO training. The loss is aggregated using `token-mean` in `verl`. For SFT and PSFT, we use a weight decay of 0.1. All experiments are conducted with full fine-tuning.

**Evaluation Benchmark.** *(i) In-domain tasks:* AIME24, AIME25, AMC, MATH-500 (Hendrycks et al., 2021), OlympidBench (He et al., 2024), and Minerva (Lewkowycz et al., 2022). *(ii) Out-of-domain tasks:* GPQA (Rein et al., 2024), ARC-C (Clark et al., 2018), TruthfulQA (Lin et al., 2021), MMLU-Pro (Wang et al., 2024), SuperGPQA (Du et al., 2025), HeadQA (Vilares & Gómez-Rodríguez, 2019) and IFEval (Zhou et al., 2023).

### C.7.1 MATH REASONING EXPERIMENTS

The detailed training configurations for SFT/PSFT are presented in Table 11.

| Method | Train batch size | Mini batch size | Learning rate | Train epochs | High clip | Cutoff_len |
|--------|------------------|-----------------|---------------|--------------|-----------|------------|
| PSFT   | 256              | 32              | 1e-6          | 10           | 0.28      | 10k        |
| SFT    | 256              | –               | 2e-5          | 10           | –         | 10k        |

Table 11: PSFT/SFT experiment configuration

### C.7.2 EXPLORATION ON MODEL POTENTIAL IN RL

The detailed training configurations for RL are presented in Table 12. For Qwen models and LLama models, we use the same configuration.

| Config | SFT + GRPO | PSFT + GRPO |
|--------|------------|-------------|
| lr | 1e-6 | 1e-6 |
| kl_coef | 0.0 | 0.0 |
| max_prompt_length | 2k | 2k |
| max_response_length | 10k | 10k |
| overlong_buffer.len | 2k | 2k |
| train_batch_size | 256 | 256 |
| ppo_mini_batch_size | 32 | 32 |
| clip_ratio_low | 0.2 | 0.2 |
| clip_ratio_high | 0.28 | 0.28 |
| temperature | 1.0 | 1.0 |
| rollout.n | 8 | 8 |
| total_training_steps | 100 | 100 |

Table 12: RL experiment configuration

### C.7.3 ALIGNMENT EXPERIMENTS

**Evaluation benchmark.** We adopt the same evaluation framework as described in Zhu et al. (2024). Since short CoT models can not solve the complex problems, we use the ARC,

GSM8K (Cobbe et al., 2021), MMLU (Hendrycks et al., 2020), GPQA, and TruthfulQA to evaluate the alignment tax. We evaluate our models on three alignment benchmarks: MT-Bench (Zheng et al., 2024a), AlpacaEval (Dubois et al., 2024), and Arena-Hard (Li et al., 2024a). We use Qwen3-30B-A3-Instruct-2507 (Yang et al., 2025) as the judge model to provide alignment evaluation. We use llm-eval-harness (Gao et al., 2024) to evaluate the alignment tax performance.

| Method | Train batch size | Mini batch size | Learning rate | Train epochs | High clip | Cutoff_len |
|---|---|---|---|---|---|---|
| PSFT | 256 | 32 | 1e-6 | 5 | 0.28 | 6k |
| PSFT$_{prolong}$ | 256 | 32 | 1e-6 | 10 | 0.28 | 6k |
| SFT | 256 | – | 2e-5 | 5 | – | 6k |

Table 13: PSFT/SFT experiment configuration

| Method | Train batch size | $\beta$ | Learning rate | Train epochs | Cutoff_len |
|---|---|---|---|---|---|
| DPO | 64 | 0.01 | 5e-7 | 1 | 4k |

Table 14: DPO experiment configuration

**Training.** The detailed training configurations for SFT/PSFT on alignment tasks are presented in Table 13. The detailed training configurations for DPO are presented in Table 14.

| Config | MT-Bench | Alpaca-Eval | Arena-Hard |
|---|---|---|---|
| temperature | 0.0 | 0.7 | 1.0 |
| top_p | – | 0.95 | 0.7 |

Table 15: Alignment benchmark generation

**Evaluation.** We adopt the settings listed in Table 15 for evaluation generation. To assess alignment tax, we use the corresponding tasks in llm-eval-harness, as summarized in Table 16.

| task | GSM8K | MMLU | TruthfulQA | CMMLU | GPQA |
|---|---|---|---|---|---|
| setting | gsm8k_cot | mmlu_flan_n_shot_generative | truthfulqa_mc1 | cmmlu | cot_n_shot |

Table 16: Alignment tax evaluation using llm-eval-harness

### C.7.4 VLM EXPERIMENTS

**Motivation.** In real-world scenarios, high-quality multimodal reasoning data is scarce, whereas large-scale textual reasoning data for LLMs is both abundant and of high quality. However, prior work has shown that directly incorporating such data can be harmful, as it may disrupt the alignment of existing MLLMs and impair their visual capabilities. Motivated by this, we design a framework in which PSFT leverages unimodal reasoning data to update only the LLM component while keeping the multimodal component fixed. We then investigate the model's capabilities after the SFT and RL stages. We hope that this pioneer exploration based on PSFT could shed light on the future multimodal reasoning learning paradigm.

**Evaluation benchmark.** *(i) In-domain text-only tasks:* AIME24, AIME25, AMC, MATH-500 (Hendrycks et al., 2021), OlympidBench (He et al., 2024), and Minerva (Lewkowycz et al., 2022). *(ii) Out-of-domain text-only tasks:* MMLU (Hendrycks et al., 2020) and MMLU-Pro (Wang et al., 2024). *(iii) In-domain text-image tasks:* Geometry-3K (Lu et al., 2021), MathVerse (Zhang et al., 2024), MATHVista (Lu et al., 2024). *(iv) Out-of-domain text-image tasks:* MMMU (Yue et al., 2024) and MMMU-Pro (Lu et al., 2024).

### C.7.5 TEXT-ONLY REASONING EXPERIMENTS

The detailed training configurations for SFT/PSFT are presented in Table 17.

### C.7.6 EXPLORATION ON MODEL POTENTIAL IN RL

The detailed RL training configurations are listed in Table 12, with the exception that the `max_response_length` is set to 5k.

| Method | Train batch size | Mini batch size | Learning rate | Train epochs | High clip | Cutoff_len |
|--------|-----------------|-----------------|---------------|--------------|-----------|------------|
| PSFT | 256 | 32 | 1e-6 | 10 | 0.28 | 6k |
| SFT | 256 | – | 2e-5 | 10 | – | 6k |

Table 17: PSFT/SFT experiment configuration

## C.8 SYSTEM PROMPT

For the reasoning task, we use the system prompt as follows: `Please reason step by step, and put your final answer within \boxed{}.`

For the alignment task, we use the system prompt as follows: `You are a helpful assistant.`

## C.9 VLM CASE STDUY

We provide two case studies, shown in Figure 16 and Figure 17, to demonstrate the feasibility of injecting text-only reasoning trajectories into VLMs.

**System Prompt ( You are a helpful assistant. )**

[Task Description]
Find $PS$
[Output]

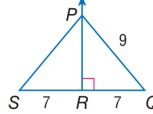

To find the length of $PS$, we start by analyzing the given information and the properties of the triangle and the right triangle formed by the altitude.
1. Identify the given lengths: - $PQ = 9$ - $SR = 7$ - $RQ = 7$
2. Since $PR$ is the altitude from $P$ to $QR$, $PR$ is perpendicular to $QR$. This makes $\triangle PRQ$ a right triangle with $\angle PRQ = 90°$.
3. The length of $PQ$ is the hypotenuse of $\triangle PRQ$. We can use the Pythagorean theorem to find the length of $PR$. The Pythagorean theorem states that in a right triangle, the square of the length of the hypotenuse (the side opposite the right angle) is equal to the sum of the squares of the lengths of the other two sides. Here, $PQ$ is the hypotenuse, and $PR$ and $RQ$ are the legs of the right triangle.

$$PQ^2 = PR^2 + RQ^2$$

Substituting the known values:

$$9^2 = PR^2 + 7^2$$

$$81 = PR^2 + 49$$

Solving for $PR^2$:

$$PR^2 = 81 - 49$$

$$PR^2 = 32$$

Taking the square root of both sides:

$$PR = \sqrt{32} = 4\sqrt{2}$$

4. Now, we need to find the length of $PS$. Since $PS$ and $PR$ are segments of the same line $PS$ and $PQ$, we can use the fact that $PS = PR + SR$.

$$PS = PR + SR$$

Substituting the known values:

$$PS = 4\sqrt{2} + 7$$

Therefore, the length of $PS$ is:

$$\boxed{4\sqrt{2} + 7} \times$$

Figure 15: Base model reasoning trajectory.

**System Prompt ( You are a helpful assistant. )**

[Task Description]
Find $PS$
Provide the answer directly without reasoning.
[Output]

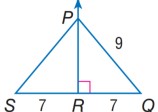

14
**Final Answer** The length $PS$ is $\boxed{14}$. $\times$

Figure 16: PSFT model without reasoning.

**System Prompt ( You are a helpful assistant. )**

[Task Description]
Find $PS$
[Output]
Okay, so I need to find the length of PS in the given triangle. Hmm.
Let's look at the image again.

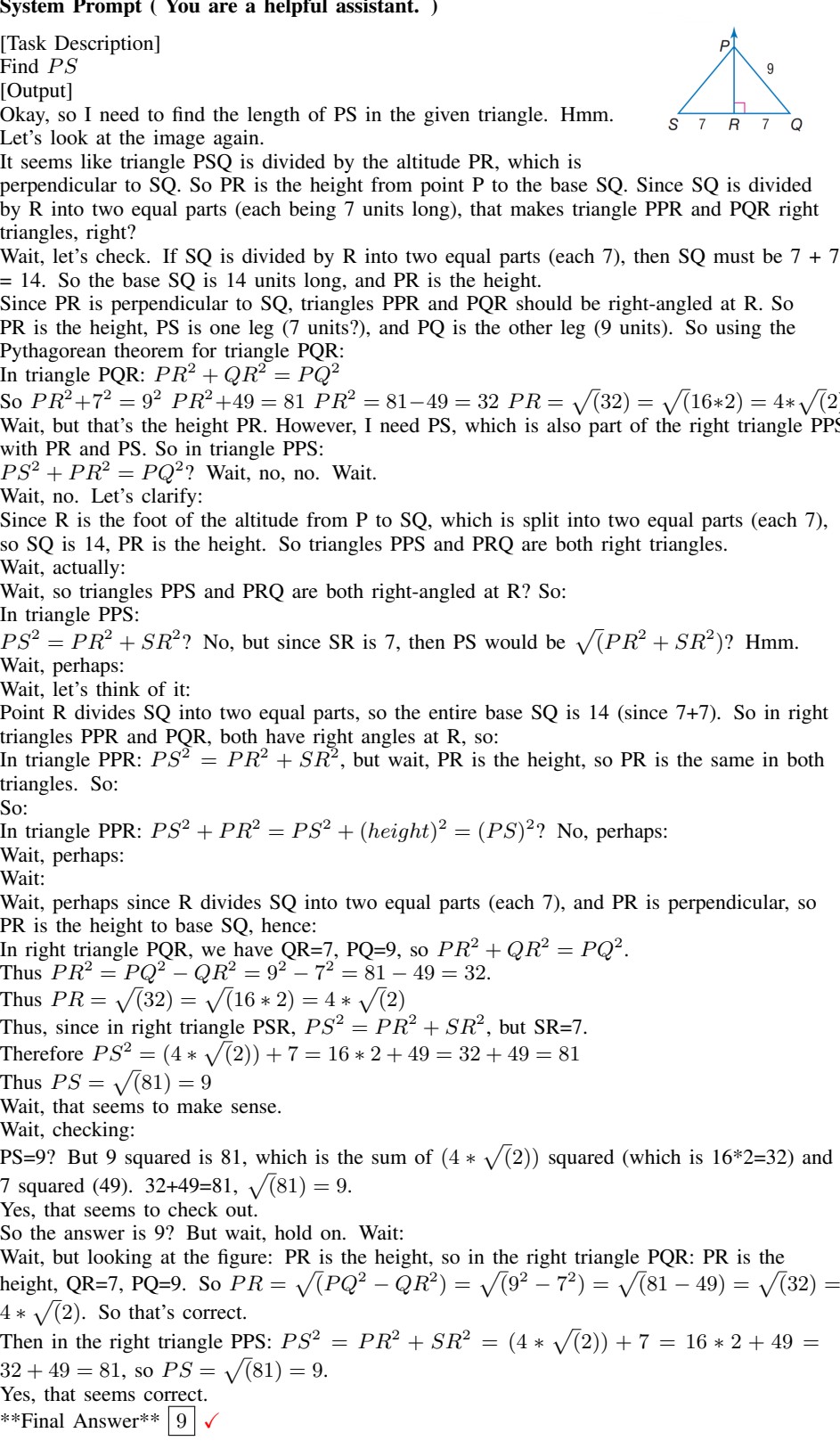

It seems like triangle PSQ is divided by the altitude PR, which is
perpendicular to SQ. So PR is the height from point P to the base SQ. Since SQ is divided
by R into two equal parts (each being 7 units long), that makes triangle PPR and PQR right
triangles, right?
Wait, let's check. If SQ is divided by R into two equal parts (each 7), then SQ must be 7 + 7
= 14. So the base SQ is 14 units long, and PR is the height.
Since PR is perpendicular to SQ, triangles PPR and PQR should be right-angled at R. So
PR is the height, PS is one leg (7 units?), and PQ is the other leg (9 units). So using the
Pythagorean theorem for triangle PQR:
In triangle PQR: $PR^2 + QR^2 = PQ^2$
So $PR^2 + 7^2 = 9^2$ $PR^2 + 49 = 81$ $PR^2 = 81 - 49 = 32$ $PR = \sqrt{(32)} = \sqrt{(16*2)} = 4*\sqrt{(2)}$
Wait, but that's the height PR. However, I need PS, which is also part of the right triangle PPS
with PR and PS. So in triangle PPS:
$PS^2 + PR^2 = PQ^2$? Wait, no, no. Wait.
Wait, no. Let's clarify:
Since R is the foot of the altitude from P to SQ, which is split into two equal parts (each 7),
so SQ is 14, PR is the height. So triangles PPS and PRQ are both right triangles.
Wait, actually:
Wait, so triangles PPS and PRQ are both right-angled at R? So:
In triangle PPS:
$PS^2 = PR^2 + SR^2$? No, but since SR is 7, then PS would be $\sqrt{(PR^2 + SR^2)}$? Hmm.
Wait, perhaps:
Wait, let's think of it:
Point R divides SQ into two equal parts, so the entire base SQ is 14 (since 7+7). So in right
triangles PPR and PQR, both have right angles at R, so:
In triangle PPR: $PS^2 = PR^2 + SR^2$, but wait, PR is the height, so PR is the same in both
triangles. So:
So:
In triangle PPR: $PS^2 + PR^2 = PS^2 + (height)^2 = (PS)^2$? No, perhaps:
Wait, perhaps:
Wait:
Wait, perhaps since R divides SQ into two equal parts (each 7), and PR is perpendicular, so
PR is the height to base SQ, hence:
In right triangle PQR, we have QR=7, PQ=9, so $PR^2 + QR^2 = PQ^2$.
Thus $PR^2 = PQ^2 - QR^2 = 9^2 - 7^2 = 81 - 49 = 32$.
Thus $PR = \sqrt{(32)} = \sqrt{(16*2)} = 4*\sqrt{(2)}$
Thus, since in right triangle PSR, $PS^2 = PR^2 + SR^2$, but SR=7.
Therefore $PS^2 = (4*\sqrt{(2)}) + 7 = 16*2 + 49 = 32 + 49 = 81$
Thus $PS = \sqrt{(81)} = 9$
Wait, that seems to make sense.
Wait, checking:
PS=9? But 9 squared is 81, which is the sum of $(4*\sqrt{(2)})$ squared (which is 16*2=32) and
7 squared (49). 32+49=81, $\sqrt{(81)} = 9$.
Yes, that seems to check out.
So the answer is 9? But wait, hold on. Wait:
Wait, but looking at the figure: PR is the height, so in the right triangle PQR: PR is the
height, QR=7, PQ=9. So $PR = \sqrt{(PQ^2 - QR^2)} = \sqrt{(9^2 - 7^2)} = \sqrt{(81 - 49)} = \sqrt{(32)} = 4*\sqrt{(2)}$. So that's correct.
Then in the right triangle PPS: $PS^2 = PR^2 + SR^2 = (4*\sqrt{(2)}) + 7 = 16*2 + 49 = 32 + 49 = 81$, so $PS = \sqrt{(81)} = 9$.
Yes, that seems correct.
**Final Answer** $\boxed{9}$ ✓

Figure 17: PSFT model with reasoning.

