# OpenReview forum: "Proximal Supervised Fine-Tuning"
_ICLR.cc/2026/Conference — ICLR 2026 Poster_

### Official Review · Reviewer_4cnt · 2025-10-31

**Soundness:** 3
**Presentation:** 3
**Contribution:** 2
**Rating:** 4
**Confidence:** 3

**Summary:**

The paper proposes Proximal Supervised Fine-Tuning (PSFT), which improves the supervised fine-tuning by utilizing a clipped surrogate objective to enforce trust-region-like constraints. The method can stabilize optimization and reduce the loss of generalization during supervised fine-tuning. Experiments demonstrate the effectiveness and statement of the method.

**Strengths:**

1. The main idea is well-motivated and straightforward.

2. The proposed method is easy to implemented with a simple objective.

3. Extensive experiments demonstrate the statements and efffectiveness of the proposed method.

**Weaknesses:**

1. Although the proposed method is well-motivated and empirically demonstrated, it seems lack of theoretical support.

2. It is not clear whether the empirical gains obtained by the proposed method or the lower effective lr and gradient clipping. It should be compared to simple baselines like: lower LR, gradient clipping, sample filtering, or reweighting.

3. The method is only compared with the common SFT and SFT SFT-KL, lack of recent RL-based fine-tuning baselines for comparisons.

4. It is not clear whether the method also work on larger model sizes and datasets.

**Questions:**

There are different conclusions in Table 1 and Table 2 for the in-domain performance, where PSFT performs better in Table 1 but not in Table 2. This also happens before and after GRPO in Table 2, where PSFT performs better than SFT after GRPO but not before. Are there any analyses or discussions on that?

---

> ### Author Response · Authors · 2025-11-18
> **Response to Reviewer 4cnt [1/2]**
>
> > [W1] Lack of theoretical support
>
> (1) Thank you for your suggestion in making PSFT theoretically and empirically complete.
>
> (2) We present the convergence result below; detailed notation and the full proof are provided in Appendix C.1.
> $$
> \frac1T\sum_{t=1}^{T}
> \mathbb E\bigl\|\nabla L^{\mathrm{PSFT}}(\theta_t)\bigr\|^{2}
> \le
> \frac{2\bigl(L^{\mathrm{PSFT}}(\theta^\ast)-L^{\mathrm{PSFT}}(\theta_1)\bigr)}
>       {\alpha\sqrt T}
> +\frac{\alpha L G^{2}(1+\epsilon)^{2}(1+\ln T)}{2T}
> $$
> When $T\to\infty$ the second term vanishes as $\mathcal O(\ln T/T)$, so the dominant rate is $\mathcal O(1/\sqrt T)$. This is consistent with the optimal upper bound of the standard non-convex SGD, indicating that under reasonable assumptions of smoothness and gradient bound, the stochastic gradient rise of PSFT has the same theoretical convergence rate as that of ordinary NLL-SFT.
>
> >  [W2] It is not clear whether the empirical gains obtained by the proposed method or the lower effective learning rate and gradient clipping.
>
> PSFT inherently incorporates gradient clipping and reweighting. As a result, increasing the learning rate improves in-domain performance while still preserving out-of-domain performance—something standard SFT cannot achieve.
>
> (1) Gradient clipping: **In fact, we conducted this experiment in the original version of the paper (Section 5.2).** A medium value of ϵ (e.g., 0.28) showing particularly favorable trade-offs between stability and performance.
>
> (2) Reweighting: **In fact, this experiment was already included in the original version of the paper (Section 5.3)**. Frequent updates deliver substantial gains on in-domain tasks.
>
> (3) Lower learning rate: We provide additional experiments in Appendix C.5. The conclusion is that increasing the learning rate helps the model overfit the training set but harms performance on other aspects. However, **thanks to our gradient clipping and reweighting mechanisms, our method generally achieves better overall performance.**
>
> | Method                             | AIME24    | AIME2025  | TruthfulQA | IFeval              |
> | ---------------------------------- | --------- | --------- | ---------- | ------------------- |
> | Base                               | 11.25     | 8.75      | 66.10      | **73.94**           |
> | SFT (lr 5e-6, bs 256)              | 21.25     | 22.50     | 66.54      | 68.87               |
> | SFT (lr 2e-5, bs 256)              | 22.08     | 23.02     | 63.14      | 54.42               |
> | PSFT (lr 1e-6, mini bs 32, bs 256) | 19.38     | 21.98     | **67.16**  | 73.03               |
> | PSFT (lr 3e-6, mini bs 32, bs 256) | **22.50** | **22.92** | 66.63      | $\underline{71.67}$ |
>
>
>
> > [W3] lack of recent RL-based fine-tuning baselines for comparisons.
>
> We provide the concurrent works iw-SFT[1] and DFT[2] for comparison on Qwen2.5-7B-Instruct using OpenR1-Math, with all settings following their respective papers. The corresponding training dynamics are presented in Appendix C.4.
>
> | Method                      | AIME24    | AIME2025  | TruthfulQA | IFeval    |
> | --------------------------- | --------- | --------- | ---------- | --------- |
> | Base                        | 11.25     | 8.75      | 66.10      | **73.94** |
> | DFT  (lr 2e-5, bs 256)      | 10.73     | 10.10     | 54.53      | 47.34     |
> | iw-SFT  (lr 2e-5, bs 256)   | 17.64     | 20.31     | 63.67      | 43.92     |
> | PSFT (lr 1e-6, mini bs 32)  | 19.38     | 21.98     | **67.16**  | 73.03     |
> | PSFT (lr 3e-6, mini bs 32 ) | **22.50** | **22.92** | 66.63      | 71.67     |
>
> **Conclusion:** PSFT has advantages in both in-domain and out-of-domain tasks.
>
> ----
>
> Reference:
>
> [1] iw-SFT: Supervised Fine Tuning on Curated Data is Reinforcement Learning (and can be improved),
>
> [2] DFT : On the Generalization of SFT: A Reinforcement Learning Perspective with Reward Rectification ,

---

> ### Author Response · Authors · 2025-11-18
> **Response to Reviewer 4cnt [2/2]**
>
> > [W4] It is not clear whether the method also works on larger model sizes and datasets.
>
> Our work focuses on SFT, and **we believe our method generalizes well.** To further demonstrate this, we additionally evaluate a 30B model on the OpenR1-Math dataset beyond the 3B, 7B, and 8B language and 7B vision–language models already tested in the paper. We also experiment with datasets of sizes 13K, 46K, and 64K, and the performance remains consistently strong across all settings. Furthermore, we believe that data quality should be of greater importance.
>
> | Qwen3-30B-A3B-Instruct-2507 | AIME24    | AIME2025  | TruthfulQA | IFeval    |
> | --------------------------- | --------- | --------- | ---------- | --------- |
> | SFT                         | 59.89     | 46.88     | 77.89      | 50.56     |
> | PSFT                        | **61.67** | **48.38** | **80.19**  | **88.52** |
>
> **Conclusion:** The results show that PSFT serves as a powerful and practical tool for industrial-scale model training.
>
> > [Q1]  PSFT performs better in Table 1 but not in Table 2.
>
> **We believe there is a misunderstanding here.**
>
> (1) PSFT with warm-up outperforms both SFT and  PSFT in in-domain performance, as shown in Table 1. In Table 2, we present results for PSFT only.
>
> (2) **As already discussed** in Section 4.2.1 (Finding 2), PSFT exhibits a slow start but rapidly catches up during the RL stage. *"Thanks to its high entropy, which promotes exploration, using PSFT as the cold start leads to a relatively steep performance improvement and ultimately surpasses the performance achieved when using the standard SFT as the cold start."*
>
>
> ----
>
> **We hope we have addressed your concern, and hope you can provide a better judgment of our work.**
>
> **Thank you!**

---

### Official Review · Reviewer_q3ok · 2025-10-31

**Soundness:** 3
**Presentation:** 4
**Contribution:** 4
**Rating:** 8
**Confidence:** 3

**Summary:**

Introduces PSFT, an algorithm to replace SFT that achieves comparable performance but archives strong generalization. It maximally preserves the model’s general capabilities (prevents entropy collapse and overfitting), and achieves good performance, especially in out-of-distribution tasks. PSFT also is compatible with RL fine-tuning afterwards, and boots RL performance by initializing with a more general model.

**Strengths:**

- Very good contribution: the PSFT algorithm is simple, yet shown to be extremely effective. I believe this is a great and timely contribution to the community. It is incredibly important to have algorithms that can fine-tune pre-trained models yet maintain their generalist performance and not overfit.
- Very clear presentation. The motivation is clear, and the preliminary covers the background work sufficiently. Explanation of the method is well done, and the results section makes the take aways crystal clear. The writing is very well done
- Comprehensive experimental evaluation shows that PSFT works very well, and evaluates on multiple base models on extensive benchmarks.

**Weaknesses:**

- The paper focuses on an algorithm that can finetune a pre-trained model with generalization ability. However, there is not much comparisons with baselines that aim to finetune a model that preserves generalization ability. The only baselines are naively doing SFT and SFT with KL constraints, and it would be nice to see comparisons with more sophisticated methods that aim to keep generalization ability.

**Questions:**

- How does PSFT compare to doing RL directly from the pre-training model without an SFT-like stage? In other words, can you outperform PSFT if you estimate the actual advantages instead of setting them all to 1.
- nit: the results plots are kind of too small to see, can you make them bigger?

---

> ### Author Response · Authors · 2025-11-18
> **Response to Reviewer q3ok**
>
> **Thank you for your strong support!**
>
> > [W1] However, there are not many comparisons with baselines that aim to fine-tune a model that preserves generalization ability.
>
> We provide the concurrent works iw-SFT[1] and DFT[2] for comparison on Qwen2.5-7B-Instruct using OpenR1-Math, with all settings following their respective papers. The corresponding training dynamics are presented in Appendix C.4.
>
> | Method                      | AIME24    | AIME2025  | TruthfulQA | IFeval    |
> | --------------------------- | --------- | --------- | ---------- | --------- |
> | Base                        | 11.25     | 8.75      | 66.10      | **73.94** |
> | DFT  (lr 2e-5, bs 256)      | 10.73     | 10.10     | 54.53      | 47.34     |
> | iw-SFT  (lr 2e-5, bs 256)   | 17.64     | 20.31     | 63.67      | 43.92     |
> | PSFT (lr 1e-6, mini bs 32)  | 19.38     | 21.98     | **67.16**  | 73.03     |
> | PSFT (lr 3e-6, mini bs 32 ) | **22.50** | **22.92** | 66.63      | 71.67     |
>
> **Conclusion:** PSFT has advantages in both in-domain and out-of-domain tasks.
>
> > [Q1] How does PSFT compare to doing RL directly from the pre-training model without an SFT-like stage? In other words, can you outperform PSFT if you estimate the actual advantages instead of setting them all to 1?
>
> (1) We have conducted this experiment before. For Qwen2.5-7B-Base, directly applying RL on the DAPO-math-17k dataset achieves an upper performance of around 21 on AIME24 (generation length 10,240). In contrast, after using SFT to inject long CoT into the base model before performing RL, the upper performance increases to around 28 on AIME24. Furthermore, zero RL is highly dependent on the pre-training and mid-training phases.
>
> (2) We build the connection between RL and SFT; therefore, we believe many RL tricks can also be applied to SFT.
>
> - In SFT, all tokens are treated as the ground truth, so all advantages are set to 1.
>
> - In PPO, advantages are estimated using a critic model and reward scores.
>
> - In GRPO, all tokens within the same trajectory share the same advantage, computed via a group relative score.
>
> - We believe your proposal is an excellent idea! Inspired by the 80/20 principle in [3], we perform SFT exclusively on these high-entropy tokens, with the advantages of other tokens estimated as 0.
>
>   | Method                     | AIME24    | AIME2025  | TruthfulQA |
>   | -------------------------- | --------- | --------- | ---------- |
>   | Qwe3-4B-PSFT               | **26.98** | **28.75** | 62.90      |
>   | Qwe3-4B-PSFT-Ent( > 0.3)   | 25.21     | 28.65     | 62.48      |
>   | Qwe3-4B-PSFT-Ent( > 0.672) | 24.17     | 25.10     | **63.71**  |
>
> **Conclusion:** We believe that this 80/20 principle update may also contribute to generalization.
>
> > [Q2] The results plots are kind of too small to see. Can you make them bigger?
>
> We apologize for this point. We intended to provide more extensive experiments to more clearly demonstrate the effectiveness of our method. With the addition of an extra page, we now have enough space to expand these figures.
>
>
> ---
>
> **All of your suggestions have been incorporated into the revised paper.**
>
> **Thank you for your support again!**
>
> References:
>
> [1] iw-SFT: Supervised Fine Tuning on Curated Data is Reinforcement Learning (and can be improved),
>
> [2] DFT : On the Generalization of SFT: A Reinforcement Learning Perspective with Reward Rectification ,
>
> [3] Beyond the 80/20 Rule: High-Entropy Minority Tokens Drive Effective Reinforcement Learning for LLM Reasoning, NeurIPS 2025.

---

> > ### Comment · Reviewer_q3ok · 2025-11-25
> >
> > Thanks for your response, it is very interesting to see the new experiments. I will keep my score.

---

> > > ### Author Response · Authors · 2025-11-26
> > >
> > > Thank you for acknowledging our new experiments.

---

### Official Review · Reviewer_UvXJ · 2025-10-31

**Soundness:** 2
**Presentation:** 3
**Contribution:** 2
**Rating:** 2
**Confidence:** 3

**Summary:**

This paper introduces a new supervised fine-tuning algorithm for LLMs that addresses the problems of overfitting and entropy collapse by incorporating trust-region updates inspired by TRPO and PPO. First, the paper reformulates the supervised fine-tuning (behavior cloning) objective in terms of the policy gradient objective and proposes an analogous clipped PPO-style objective.
The paper evaluates the proposed algorithm on several math reasoning benchmarks and investigates its effectiveness for RL post-training, human alignment, and vision-language models.

**Strengths:**

- The paper is clear and easy to follow.
- The proposed method is simple, and the results seem promising.

**Weaknesses:**

- The analogy drawn between the proposed PSTF objective and TRPO/PPO-style objectives may be oversimplified.

- The paper lacks comparisons with stronger supervised fine-tuning baselines.

- The evaluation is primarily limited to mathematical reasoning benchmarks.

**Questions:**

- The analogy between PSFT and RPO/PPO-like objectives is not accurate: (1) In the ppo objective in equation 5 the expectation is taken w.r.t the old policy $\pi_old$, while in the PSFT objective, the expectation is w.r.t the offline dataset. (2) The probability ratio in PPO and PSFT loss is $\frac{\pi(a \mid s)}{\pi_{old}(a \mid s)}$, but it should something like  $\frac{\pi(a \mid s)}{\pi_{D}(a \mid s)}$.
Can the authors comment on these differences and how they would explain the performance gains?
- The method is compared with two baselines, vanilla SFT and SFT with KL. Are there any other SFT baselines that the authors can compare to?
- Performing an SFT warm-up seems to contradict the paper’s stated goal of replacing vanilla SFT with a more stable method that prevents entropy collapse. The paper claims that the warm-up stage helps mitigate the bias resulting from the misalignment between the data distribution and πold\pi_{\text{old}}πold​, but this warm-up phase may also affect some of the model’s capabilities. It is not clear how this aligns with the core motivation of the proposed approach.
- Most of the evaluation is on Math benchmarks; can the authors show some results on coding or general reasoning tasks?

I am willing to increase my score if the authors addressed my concerns.

---

> ### Author Response · Authors · 2025-11-18
> **Response to Reviewer UvXJ**
>
> > [W1 & Q1] The analogy between PSFT and TRPO/PPO-like objectives is not accurate
>
> (1) TRPO/PPO is introduced to incorporate a **soft trust region**, which we then apply as a constraint to the supervised learning objective. This mechanism **enables stable optimization on the target domain while preserving out-of-domain generalization**.
>
> (2) The trust region is computed between $\pi_{\theta}$ and $\pi_{\text{old}}$, where $\pi_{\text{old}}$ is dynamically updated in PPO. We preserve this property in PSFT, which is **key to the observed performance gains** (see Section 5.3). Specifically, $\pi_{\text{old}}$ is initialized from the reference model $\pi_{\text{ref}}$, after training begins, $\pi_{\text{old}}$ is updated to the previous $\pi_\theta$, thereby enabling continual improvements.
>
> (3)  Indeed, we acknowledge that their forms are quite similar. However, we explicitly highlight the differences between the PSFT and PPO objectives; our discussion may not constitute an “analogy” but rather a **comparison**. For example:
>
> - Translating this idea to the supervised setting with an offline dataset $\mathcal{D}$,
> - $L^{\mathrm{PSFT}}(\theta)=\mathbb{E}_{(s_t, a_t) \sim \mathcal{D}}[\dots]$,
> - Setting $A = 1$, which makes it similar to standard SFT.
>
> > [W2 & Q2] Are there any other SFT baselines that the authors can compare to?
>
> We provide the concurrent works iw-SFT[1] and DFT[2] for comparison on Qwen2.5-7B-Instruct using OpenR1-Math, with all settings following their respective papers. The corresponding training dynamics are presented in Appendix C.4.
>
> | Method  | AIME24    | AIME2025  | TruthfulQA | IFeval    |
> | --------------------------- | --------- | --------- | ---------- | --------- |
> | Base   | 11.25    | 8.75 | 66.10  | **73.94** |
> | DFT  (lr 2e-5, bs 256)      | 10.73   | 10.10   | 54.53  | 47.34     |
> | iw-SFT  (lr 2e-5, bs 256)   | 17.64     | 20.31     | 63.67      | 43.92     |
> | PSFT (lr 1e-6, mini bs 32)  | 19.38     | 21.98     | **67.16**  | 73.03     |
> | PSFT (lr 3e-6, mini bs 32 ) | **22.50** | **22.92** | 66.63      | 71.67     |
>
> **Conclusion:** PSFT has advantages in both in-domain and out-of-domain tasks.
>
> > [Q3 ] Performing an SFT warm-up seems to contradict the paper’s stated goal of replacing vanilla SFT with a more stable method that prevents entropy collapse.
>
> (1) In our original paper, we described the warm-up phase using the term **“optionally”**. Including the warm-up is intended to improve target task performance, as stated in the paper.
>
> (2) As shown in Figure 1, PSFT with warm-up still effectively prevents entropy collapse compared to SFT alone.
>
> (3) You mentioned that *“The paper claims that the warm-up stage helps mitigate the bias resulting from the misalignment between the data distribution and πold, ”*; $\rightarrow$ However, as mentioned in the preceding sentence, **allowing the old policy $\pi_{\text{old}}$ to evolve dynamically is essential.**
>
> (4) You mentioned that *"but this warm-up phase may also affect some of the model’s capabilities."* $\rightarrow$ **Yes, directly applying PSFT is sufficient**, whereas SFT exhibits certain limitations.
>
> > [W3 & Q4] Can the authors show some results on coding or general reasoning tasks?
>
> (1) **We clarify that PSFT has been applied to multiple domains, including math reasoning, long-chain-of-thought data, dialogue (Section 4.3), and multi-modal tasks (Section 4.4). The differences across these domains are substantial**, demonstrating the effectiveness of PSFT.
>
> (2) We now include code-related experiments on Qwen2.5-7B-Instruct as follows: (i) For code SFT training, the training format is *“Please reason step by step and return your code.”* + Question + Code.   (ii) For math evaluation, we use the prompt: *“Please reason step by step and put your answer in a box.”*
>
> | SFT | Step 100 |  200 | 300 |  400 | 500 |
> | ---------------------- | -------- | --------- | -------- | -------- | -------- |
> | Coding (LiveCodeBench) | 23.89 | 27.96| 26.45 |26.54| 27.11    |
> | Math (AIME-24) | 0.03  | 0.05 | 0.03     | 0.06  | 0.06  |
>
> | PSFT  | Step 100  |  200  | 300  |400  |  500  |
> | ---------------------- | --------- | --------- | --------- | --------- | --------- |
> | Coding (LiveCodeBench) | 20.67 | 27.20 | 27.96| 28.15| 28.53|
> | Math (AIME-24) | **13.44** | **14.58** | **15.52** | **16.46** | **16.77** |
>
> **Conclusion:** These results show that while both SFT and PSFT effectively learn the target domain, PSFT better preserves mathematical instruction-following ability.  **Now, we validate PSFT across math, code, dialogue, and multi-modal tasks, all of which demonstrate strong performance.**
>
> ---
>
> **We hope we have addressed your concern, and hope you can provide a better judgment of our work.**
>
> **Thank you!**
>
> Reference:
>
> [1] iw-SFT: Supervised Fine Tuning on Curated Data is Reinforcement Learning (and can be improved),
>
> [2] DFT : On the Generalization of SFT: A Reinforcement Learning Perspective with Reward Rectification ,

---

> > ### Comment · Reviewer_UvXJ · 2025-11-26
> >
> > Dear authors,
> >
> > Thank you for the clarifications and the new experiments.
> > Most of the responses address my concern. I think it is better to formulate the discussion regarding PPO as a comparison to avoid theoretical confusion.
> >
> >  The new empirical results are more convincing, and I like the method's simplicity.
> > Best.

---

> > > ### Author Response · Authors · 2025-11-26
> > >
> > > Thank you for acknowledging our work. Following your suggestion, we have added an explicit comparison with PPO in Section 3 of the revised paper. We will continue to refine the clarity and presentation of this section in future revisions.
> > >
> > > This is truly the best news I’ve received recently!

---

### Official Review · Reviewer_Z7c6 · 2025-11-01

**Soundness:** 3
**Presentation:** 3
**Contribution:** 2
**Rating:** 4
**Confidence:** 3

**Summary:**

The paper introduces Proximal Supervised Fine-tuning, a method for supervised fine-tuning that takes inspiration from trust-region policy optimisation (TRPO) and proximal policy optimisation (PPO) from reinforcement learning. The supervised fine tuning setting is matched to the RL context by setting the advantage function to be a constant function and matching the objectives. Then the importance sampling of TRPO and clipping of PPO are applied. Experiments are carried out on a wide number of domains and applications.

The novelty of this paper is in an interesting composition of existing methods.

**Strengths:**

- The paper presents a simple yet effective idea.
- Generally well written, except for the points mentioned below.
- Good set of experiments with clear descriptions of findings.
- Code provided + open-sourced datasets and models.

**Weaknesses:**

- There are inconsistencies in the description of the mathematical notation:
	- - Section 2: a MDP usually has a reward function as well with respect to which the *best decision* is made.  You have omitted this. Why?
	- Line 78 onwards: What does the $\*$ in $a^\*_t$ signify: the optimal action? If so, state this in the text. What is the distinction between the action in Equation 2 and 3, where Equation 2 contains the $\*$ and Equation 3 does not?
	- Line 83: The authors should define what an advantage is, at least in text.
- References are missing at places (below are some examples):
	- Line 84: Please add a reference after "policy gradient theorem".
	- Line 105: "... can be difficult to optimize in practice." — please state why and provide a reference.
	- Line 115: "... preventing entropy collapse..." — please say what entropy collapse is and provide a reference.
	- Line 257: "... DAPO ..." — please add a reference to DAPO.
		- Can you elaborate on why DAPO was chosen?
- I am wondering if the framing of this paper can be improved / made more accurate by changing it from applying Reinforcement Learning to Supervised Fine Tuning to applying the specific techniques (importance sampling and clipping) to the supervised objective, and observing / realising similar benefits. In particular, because the advantage is set to a constant 1, I am hesitant to classify the presented work as RL.

### Suggestions
- Line 27: Please expand PPO and GRPO here since this is the first time these abbreviations are used.
- Line 31: "These reasoning models offer an abundant and valuable latent thoughts (Ruan et al., 2025) across the internet" — It is unclear to me what this sentence means. Are the authors saying that foundation models are used to generate the thought data?
- Line 68: "the state space (partial sequences), ..." — Elaborate on what you mean by "partial sequences", since it is unclear here. E.g., partial sequences of what?
- Line 115: "... while retaining general capabilities..." — can you expand on the specific capabilities you are targeting?

**Questions:**

- Line 58: "It maximally preserves..." — Do you have a proof to supporting the claim that your method "*maximally* preserves the model's general capabilities"?
- Line 102: "... far from the reference policy, enabling more targeted and ..." — isn't the reference policy $\pi_{\theta_\mathrm{old}}$? If so, is this the policy from the previous update step? If this is the case, is calling it a reference policy valid, since reference policy implies a notion of a ground-truth. Furthermore, if this is the policy from the previous step, how does this method ensure the training to be "more targeted" — what is the target if the target keeps changing at each step?
	- Alternatively, does $\pi_{\theta_\mathrm{old}}$ refer to the initial policy? If so, couldn't the notation be changed to $\pi_{\theta_0}$, or something else to that effect?

---

> ### Author Response · Authors · 2025-11-18
> **Response to Reviewer  Z7c6**
>
> > [W1 ]  There are inconsistencies in the description of the mathematical notation
>
> (1) Since the reward function $R$ is not used later, we omit it for simplicity. However, we note that this is not strictly accurate, and have accordingly revised the paper.
>
> (2) $a^*_t$ denotes the optimal action. In contrast, Equation 3 involves RL sampling, which may not yield the optimal action.
>
> (3) We have added a description of the advantages in the revised paper.
>
> > [W2] References are missing at places
>
> (1) $\rightarrow$  policy gradient theorem (Sutton et al., 2000)
>
> (2) $\rightarrow$ " can be difficult to optimize in practice because of complex second-order optimization and sensitivity to the constraint threshold (Schulman et al., 2017)."
>
> (3) $\rightarrow$ preventing entropy collapse (the over-concentration of the output distribution that reduces diversity and harms generalization (Ziegler et al., 2019)),
>
> (4) $\rightarrow$  DAPO is GRPO with four tricks (clip higher, overlong penalty, resample, and token-level loss) to stable RL training. (Yu et al., 2025)
>
> > [W3 ] I am hesitant to classify the presented work as RL.
>
> ***Proximal Supervised Fine-tuning (PSFT)* is an improved algorithm at the SFT stage.** We have mentioned this in various places throughout the paper (such as the Title, abstract, contribution of introduction). Introducing TRPO/PPO is to introduce concepts such as importance sampling (Line 100, TRPO) and the soft trust region (Line 114, PPO). We then explain how these ideas can be translated to the supervised learning setting using an offline dataset (Section 3). **We believe this framing is consistent with the perspective you proposed.** The RL part of our experiment is to verify that the advantages of our PSFT in the SFT stage can also continue to influence the further RL stage.
>
> > [Suggestion]
>
> (1) We have expanded the abbreviation in the revised paper.
>
> (2) ''These reasoning models offer an abundant and valuable latent thoughts (Ruan et al., 2025) across the internet'' $\rightarrow$ "producing higher-quality reasoning trajectories."
>
> (3) "partial sequences" $\rightarrow$  "the prefix of the current decoding step"
>
> (4) "while retaining general capabilities"  $\rightarrow$ "while retaining general capabilities beyond the target domain—such as scientific reasoning and instruction following"
>
> > [Q1]  Do you have a proof to support the claim that your method "*maximally* preserves the model's general capabilities"?
>
> (1) This conclusion is supported by our experimental results. We believe that OOD tasks can, to some extent, reflect the general capabilities acquired during pre-training. For example, after learning long chain-of-thought mathematical reasoning, we evaluate OOD tasks such as GPQA and IFEval, and observe that PSFT consistently outperforms standard SFT.
>
> (2) **We have adjusted it to the comparative form.**
>
> $\rightarrow$  “Compared to standard SFT, PSFT preserves the model’s general capabilities while achieving comparable performance on target tasks.”
>
> > [Q2] Misinterpretation of TRPO
>
> We apologize for previously using the two terms, “reference model” and “targeted,” inaccurately when introducing TRPO. The use of “reference model” was our mistake. We used “targeted” to indicate the direction toward the “reward.”
>
> **We have revised the description to correct this issue.**
>
> $\rightarrow$  TRPO ensures that each policy update stays within a trust region by enforcing a hard KL constraint, preventing the new policy from deviating too far from the previous one and thus maintaining stable and reliable learning.
>
>
>
> **Thank you for your detailed, insightful, and invaluable suggestions for polishing our paper.**  All revisions have been highlighted in blue.
>
> **We hope this revised version meets your expectations, and we would greatly appreciate it if you could reconsider your evaluation.**
>
> ---
>
> References:
>
> [1] Richard S. Sutton, David McAllester, Satinder Singh, and Yishay Mansour. Policy gradient methods for reinforcement learning with function approximation. In Advances in Neural Information Processing Systems (NeurIPS), volume 12, 2000.
>
> [2] John Schulman, Filip Wolski, Prafulla Dhariwal, Alec Radford, and Oleg Klimov. Proximal policy optimization algorithms. arXiv preprint arXiv:1707.06347, 2017.
>
> [3] Daniel M. Ziegler, Nisan Stiennon, Jeffrey Wu, Tom B. Brown, Alec Radford, Dario Amodei, Paul F. Christiano, and Geoffrey Irving. Fine-tuning language models from human preferences. arXiv preprint arXiv:1909.08593, 2019.
>
> [4] Qiying Yu, Zheng Zhang, Ruofei Zhu, Yufeng Yuan, Xiaochen Zuo, Yu Yue, Weinan Dai, Tiantian Fan, Gaohong Liu, Lingjun Liu, et al. Dapo: An open-source llm reinforcement learning system at scale. arXiv preprint arXiv:2503.14476, 2025.

---

### Official Review · Reviewer_XwCA · 2025-11-11

**Soundness:** 3
**Presentation:** 2
**Contribution:** 2
**Rating:** 2
**Confidence:** 3

**Summary:**

This paper proposes a novel supervised fine-tuning method called Proximal Supervised Fine-Tuning (PSFT), which aims to address the issues of degraded generalization, entropy collapse, and excessive policy updates commonly found in traditional SFT. Inspired by TRPO and PPO from reinforcement learning, PSFT introduces a clipped probability ratio to constrain the magnitude of policy updates. The authors conduct extensive experiments across various domains, showing that PSFT achieves comparable in-domain performance to SFT while offering advantages in out-of-domain generalization and entropy stability.

**Strengths:**

- Originality: The paper introduces the trust-region mechanism from reinforcement learning into supervised fine-tuning, proposing PSFT. This is a novel and reasonable perspective that offers fresh insights into improving SFT.
- Quality: The authors validate the effectiveness of the proposed method through extensive experiments and compare it with several existing methods. Results show consistent performance improvements across multiple benchmarks.
- Clarity: The structure of the paper is generally clear, although there are some notational errors that need correction.

**Weaknesses:**

1. Although PSFT is inspired by TRPO/PPO, it remains largely heuristic, lacking rigorous mathematical analysis on convergence, stability, or generalization error.
2. While PSFT provides a better cold-start for RL, the RL stage still uses standard GRPO/DAPO. The paper does not explore joint optimization of PSFT and RL objectives.
3. All experiments are conducted on 7B–8B scale models. The effectiveness of PSFT on larger models (e.g., 30B+) remains unverified.
4. Current experiments mainly focus on mathematical reasoning and long CoT data. It is unclear whether PSFT is also applicable to dialogue, code, instruction following, or other general SFT tasks.
5. Equation (3) is incorrect — the objective function is not in the right form.
6. The assumption $\hat{A}_t=1$ is too strong. The paper does not justify why this assumption is reasonable.
7. In Equation (7), the meaning of $r_t$ is not clearly explained.
8. The meaning of $π_{old}$ is ambiguous. It is unclear why the same \*π\*old is used throughout training on a fixed dataset, and thus the effectiveness of this approach remains questionable.

**Questions:**

See Weaknesses.

---

> ### Author Response · Authors · 2025-11-18
> **Response to Reviewer XwCA [1/2]**
>
> > [W1] Lacking rigorous mathematical analysis on convergence, stability, or generalization error.
>
> (1) Thank you for your suggestion in making PSFT theoretically and empirically complete.
>
> (2) We present the convergence result below; detailed notation and the full proof are provided in Appendix C.1.
> $$
> \frac1T\sum_{t=1}^{T}
> \mathbb E\bigl\|\nabla L^{\mathrm{PSFT}}(\theta_t)\bigr\|^{2}
> \le
> \frac{2\bigl(L^{\mathrm{PSFT}}(\theta^\ast)-L^{\mathrm{PSFT}}(\theta_1)\bigr)}
>       {\alpha\sqrt T}
> +\frac{\alpha L G^{2}(1+\epsilon)^{2}(1+\ln T)}{2T}
> $$
> When $T\to\infty$ the second term vanishes as $\mathcal O(\ln T/T)$, so the dominant rate is $\mathcal O(1/\sqrt T)$. This is consistent with the optimal upper bound of the standard non-convex SGD, indicating that under reasonable assumptions of smoothness and gradient bound, the stochastic gradient rise of PSFT has the same theoretical convergence rate as that of ordinary NLL-SFT.
>
> > [W2] The paper does not explore joint optimization of PSFT and RL objectives.
>
> (1) Our work proposes an improved SFT that can acquire the target capability while preserving general abilities, paving the way for further optimization.  We emphasize that staged training remains the most effective and widely adopted approach.
>
> (2) We consider joint optimization as a separate line of work, independent of our research. We highlight a feasible direction for future exploration: *performing PSFT while simultaneously sampling negative examples during RL, enabling an RL procedure akin to 2-GRPO.*
>
> > [W3] All experiments are conducted on 7B–8B scale models. larger models (e.g., 30B+) remain unverified
>
> **We clarify that our experiments also included models at the 4B scale.** To further strengthen our evidence, we additionally report results on the Qwen-30B-A3B MoE model trained on the OpenR1-Math dataset beyond the 3B, 7B, and 8B language and 7B vision–language models already tested in the paper.
>
> | Qwen3-30B-A3B-Instruct-2507 | AIME24    | AIME2025  | TruthfulQA | IFeval    |
> | --------------------------- | --------- | --------- | ---------- | --------- |
> | SFT                         | 59.89     | 46.88     | 77.89      | 50.56     |
> | PSFT                        | **61.67** | **48.38** | **80.19**  | **88.52** |
>
> Conclusion: **The results show that PSFT serves as a powerful and practical tool for industrial-scale model training.** Training dynamic see Appendix C.3.
>
> ---
> References:
>
> [1] 2-GRPO: It Takes Two: Your GRPO Is Secretly DPO, https://arxiv.org/abs/2510.00977

---

> ### Author Response · Authors · 2025-11-18
> **Response to Reviewer XwCA [2/2]**
>
> > [W4] It is unclear whether PSFT is also applicable to dialogue, code, instruction following, or other general SFT tasks.
>
> (1) **We clarify that PSFT has been applied to multiple domains, including math reasoning, long-chain-of-thought data, dialogue (Section 4.3), and multi-modal tasks (Section 4.4). The differences across these domains are substantial** , demonstrating the effectiveness of PSFT.
>
> (2) We now include code-related experiments on Qwen2.5-7B-Instruct as follows: (i) For code SFT training, the training format is *“Please reason step by step and return your code.”* + Question + Code.  (ii) For math evaluation, we use the prompt: *“Please reason step by step and put your answer in a box.”*
>
> | SFT                    | Step 100 | Step 200 | Step 300 | Step 400 | Step 500 |
> | ---------------------- | -------- | -------- | -------- | -------- | -------- |
> | Code (LiveCodeBench) | 23.89    | 27.96    | 26.45    | 26.54    | 27.11    |
> | Math (AIME-24)         | 0.03     | 0.05     | 0.03     | 0.06     | 0.06     |
>
> | PSFT                   | Step 100  | Step 200  | Step 300  | Step 400  | Step 500  |
> | ---------------------- | --------- | --------- | --------- | --------- | --------- |
> | Code (LiveCodeBench) | 20.67     | 27.20     | 27.96     | 28.15     | 28.53     |
> | Math (AIME-24)         | **13.44** | **14.58** | **15.52** | **16.46** | **16.77** |
>
> **Conclusion:** These results show that while both SFT and PSFT effectively learn the target domain, PSFT better preserves mathematical instruction-following ability. **Now, we validate PSFT across math, code, dialogue, and multi-modal tasks, all of which demonstrate strong performance.**
>
> > 【W5】Equation (3) is incorrect
>
> RL performs policy gradient ascent, and therefore, we preserve this objective.  To avoid confusion, we clarify the maximization of this objective in the revised paper in Section 2.1 (highlighted in blue).
>
> > [W6] The assumption $\hat A_t = 1$ is too strong. The paper does not justify why this assumption is reasonable.
>
> (1) In SFT, all tokens are treated as the ground truth, so all advantages are set to 1. Section 2.1 justifies this assumption.
>
> (2) **PSFT is an improved variant of SFT.** Therefore, we adhere to the SFT advantages estimation. Exploring more fine-grained offline advantages could be a promising direction for future work.
>
> > [W7]  the meaning of $r_t$ is not clearly explained.
>
> Thank you for pointing this out. It should be $r_t(\theta)$, and we have updated the paper accordingly (highlighted in blue).
>
> > [W8] It is unclear why the same $\pi_{\text{old}}$ is used throughout training on a fixed dataset
>
> (1) We follow the standard notation used in PPO. $\pi_{\text{old}}$ is dynamically updated.
>
> (2) In Section 3, **we clarify that we have pointed out  $\pi_{\text{old}}$ is dynamically updated during training**. In Section 5.3 (Table 6), we conducted an ablation study showing that using a fixed $\pi_{\text{old}}$ limits target performance, and we further investigate how the update frequency affects performance.
>
> (3) To avoid confusion, we introduce the notation $\pi_{\theta_{\text{ref}}}$ to explicitly refer to a fixed $\pi_{\text{old}}$ in Section 3 (highlighted in blue).
>
> **We hope we have addressed your concern, and hope you can provide a better judgment of our work.**
>
> **Thank you!**

---

### Author Response · Authors · 2025-12-02
**Final Summary [1/2]**

Dear ACs/SACs/PCs

**We sincerely thank you for your time and effort in managing the review process for our paper.** We provide a summary of the current status of our paper and the progress of our rebuttal.

## 1. Summary of Contributions

   PSFT introduces a clipped surrogate objective to stabilize SFT updates, preventing entropy collapse and overfitting while preserving generalization, and is **extensively validated across models and tasks** as a strong alternative to standard SFT, better supporting robust downstream RL.

## 2. Reviewer Status

Below, we summarize the current status:

| Reviewer | Strengths                                                    | Weaknesses / Questions                                       | Attitude                                                    |
| -------- | ------------------------------------------------------------ | ------------------------------------------------------------ | ----------------------------------------------------------- |
| Q3ok     | Very good contribution; very clear presentation; comprehensive experimental evaluation | Lack of baselines; small plots; PSFT future work             | **Maintained the positive score of 8.**                     |
| UvXJ     | Clear presentation; simple yet effective method              | PPO vs. PSFT comparison; lack of baselines; PSFT warm-up necessity; applicability to other tasks | **Acknowledged our response and raised score from 2 to 8.** |
| Z7c6     | Simple yet effective; good writing; solid experiments; code provided | Paper writing and presentation                               | --                                                          |
| 4cnt     | Clear motivation; easy to implement; extensive experiments   | Lack of theoretical analysis; learning rate a blation; more baselines; larger models | --                                                          |
| XwCA     | Novel and reasonable; extensive experiments; clear structure | Lack of theoretical analysis; lack of larger-model results; applicability to other tasks; notation problems | --                                                          |

**We believe we have addressed all of the above concerns and made the necessary revisions.**

## 3. Revision Summary

**All the reviewers acknowledge that our work is well-motivated, straightforward, supported by extensive experiments, clearly written, and novel.** To address the reviewers’ concerns, we have made the following revisions:

（1）Added theoretical analysis. (Reviewers `XwCA`, `4cnt`)

（2）Added experiments on a 30B model (now including 4B, 7B, 8B, 30B, and a 7B vision LLM). (Reviewers `XwCA`, `4cnt`)

（3）Added DFT and iw-SFT baseline experiments (now including SFT, SFT-KL, DFT, and iw-SFT). (Reviewers `UVXJ`, `4cnt`)

（4）Added code-task experiments (now covering math, dialogue, code, long-CoT, and multimodal tasks). (Reviewers `XwCA`, `UVXJ`, `q3ok`)

（5）Added learning-rate ablation experiments. (Reviewer `4cnt`)

（6）Added discussion on derivative directions based on PSFT. (Reviewers `q3ok`, `XwCA`)

（7）Added a comparison between PPO and PSFT. (Reviewers `Z7c6`, `UVXJ`)

---

> ### Author Response · Authors · 2025-12-02
> **Final Summary [2/2]**
>
> **Before the OpenReview information leak incident**, we had already received responses from two reviewers:
>
> - **Reviewer UvXJ** acknowledged that we addressed most of his concerns and **raised his score from 2 to 8**.
> - **Reviewer q3ok** acknowledged our response and maintained his positive score of **8**.
>
>
> In the following part, we provide clarifications regarding the remaining concerns. We would appreciate it if the AC could assist us by reviewing the following reviewers in more detail.
>
> ------
>
> ### **Reviewer XwCA**
>
> Reviewer XwCA praises our work as novel and well-motivated, with extensive experiments, consistent performance, and clear writing. However, we do not agree with some of the weaknesses pointed out. Nevertheless, we still supplemented the additional experiments required by the reviewer XwCA to demonstrate our validity.
>
> **1. Oversights:**
>
> - *“All experiments are conducted on 7B–8B scale models.”*
>    → In fact, we also evaluate PSFT on a **4B model** and a **7B vision-language model** with a vision encoder. Furthermore, we added **30B models during the rebuttal**.
> - *“Current experiments mainly focus on mathematical reasoning and long CoT data. It is unclear whether PSFT is also applicable to dialogue, code, instruction following, or other general SFT tasks.”*
>    → We also evaluate PSFT on **dialogue** and **multimodal** datasets, as described in Sections 4.3 and 4.4. Additionally, we included **code tasks during the rebuttal**.
> - *“It is unclear why the same $\pi_{old}$ is used throughout training on a fixed dataset.”*
>    → As stated in lines 129–132 of our paper, **$\pi_{old}$ is dynamically updated** during training. Additionally, Section 5.3 of the original paper presents experiments conducted specifically to demonstrate this dynamic update.
>
> **2. Misunderstandings:**
>
> - *“Equation (3) is incorrect.”*
>    → This appears to arise from a misunderstanding of **policy gradient ascent in RL**, leading to a misinterpretation. Equation (3) is not a negative optimization; we have explicitly clarified this in the paper.
> - *“The assumption $A_t = 1$ is too strong.”*
>    → This overlooks that our work treats **SFT as a special case of RL**, where this assumption naturally holds. Specifically, we set the advantage to 1 following the standard SFT formulation.
>
> **3. Unreasonable demands:**
>
> - *“The paper does not explore joint optimization of PSFT and RL objectives.”*
>    → Our method is designed as an **optimized SFT algorithm**, whereas the reviewer’s comment concerns a **combined PSFT–RL joint optimization**, which is beyond the paper’s intended scope. To demonstrate that PSFT leaves room for RL optimization, **we had included one section performing first PSFT and then RL**. However, the reviewer’s remark that *"the RL stage still uses standard GRPO/DAPO"* and the suggestion to provide a joint optimization of both, which is not directly related to our work. Additionally, it is worth noting that performing SFT first, followed by RL, remains the most commonly adopted approach in industry.
>
> Furthermore, regarding reviewer **XwCA**, whose review was submitted on **November 11**, 11 days past the required deadline.
>
> We still supplemented the vast majority of the experiments as requested by the reviewers as much as possible, and gave full explanations to the reviewers' questions and potential misunderstandings. **We hope the AC can consider this reviewer XwCA’s comments with appropriate care.**
>
> ------
>
> ### **Reviewer Z7c6**
>
> Reviewer Z7c6 praises our work as simple and effective, well-written, with detailed experimental explanations and provided code. **Most of the comments are on writing issues**, and we have addressed every detail as shown in the **blue text** in our revised version.
>
> ------
>
> ### **Reviewer 4cnt**
>
> Reviewer 4cnt praises our work as well-motivated, straightforward, easy to implement, and supported by extensive experiments. To address this reviewer’s concerns, we provide theoretical analysis, a learning rate ablation study, additional baselines, and code tasks to further validate PSFT’s effectiveness. **Regarding the sole question raised by this reviewer**, we believe it stems from a **careless reading**: the results in Table 1 and Table 2 are consistent. PSFT outperforms SFT after GRPO due to its higher entropy, which encourages better exploration. Using PSFT as the cold start leads to a steeper initial improvement and ultimately surpasses the performance achieved when using standard SFT as the cold start, as already discussed in the paper. This is also one of the benefits of our method.
>
> ------
>
> We sincerely thank the ACs and PCs for their time and careful consideration of our rebuttal. We hope our clarifications and additional experiments help demonstrate the contributions and validity of our work. **We greatly appreciate your guidance and support throughout this process.**

---

### Meta-Review · Area_Chair_9peR · 2025-12-14

**Summary:**

The paper proposes Proximal Supervised Fine-Tuning (PSFT), a fine-tuning objective that views Supervised Fine-Tuning (SFT) as a special case of a policy gradient method with constant positive advantages. PSFT incorporates a trust-region-inspired, clipped surrogate objective, drawing inspiration from TRPO and PPO in RL. The method aims to prevent issues like degraded generalization and entropy collapse often seen in standard SFT. Experiments across mathematical, human-value, code, dialogue, and multimodal domains show that PSFT matches standard SFT in-domain, improves OOD generalization, and provides a foundation for subsequent RL optimization.

The initial reviews were partially positive, acknowledging the work as novel, well-motivated amd simple yet effective. However, primary concerns were raised, including (1) a lack of rigorous theoretical analysis on convergence, stability, or generalization error, (2) insufficient evidence for its effectiveness on larger models (e.g., 30B+), though I think this is beyond available computation in academia , (3) limited comparison with other sophisticated SFT baselines that also aim to preserve generalization, and (4) questions regarding its applicability to diverse tasks beyond math reasoning and long CoT data. There were also several notational and conceptual clarification issues.

During the rebuttal phase, the authors largely addressed these concerns through substantial revisions and supplementary empirical analysis. They provided additional theoretical convergence analysis, which I suggest the author to double check for some inequalities. They added experiments on a 30B model (Qwen3-30B-A3B)  and extended the evaluations to include code tasks and comparisons with DFT and iw-SFT baselines. After checking the updated manuscript, I believe the newly added explanation and experiments strengthen the manuscript. It is necessary to incorporate all newly added results and theoretical analysis from the rebuttal into the final version of the manuscript. Given the above, I suggest an acceptance.

**Reviewer Concerns:**

### Addressed Concerns

(1) Lack of theoretical support: The authors provided a theoretical convergence result for PSFT.
(2) Limited model size evaluation: Experiments were added on a 30B model , showing that PSFT is effective at industrial scale.
(3) Lack of baselines: The authors added comparisons against DFT and iw-SFT, which are concurrent, RL-based fine-tuning baselines.
(4) Limited task applicability: The authors clarified that dialogue and multimodal tasks were in the original paper and added new experiments on code-related tasks (LiveCodeBench).
(5) Notation and conceptual clarification: Issues regarding the dynamic update of $\pi_{old}$, the justification for the $\hat{A}_t=1$ advantage assumption in SFT and minor notation errors were addressed.

### Remaining Concerns

No new, major concerns were introduced. The one-off concerns about small plots and the a priori need for certain experiments were resolved by the authors' extensive additions. The primary remaining task is for the authors to ensure all the substantial new theoretical analysis and empirical results from the rebuttal (especially the 30B, code, and new baseline results) are fully and clearly integrated into the final manuscript.

**Reviewer Scores:**

Reviewer UvXJ explicitly raised their score from 2 to 8 after the rebuttal

---

### Decision · Program_Chairs · 2026-01-26

Accept (Poster)